# A generalizable assay for intracellular accumulation to profile cytosolic drug delivery in mammalian cells

Sobika Bhandari[1], George M. Ongwae[1], Rachita Dash[1], Zichen Liu[1], Mahendra D. Chordia[1], Yuchen He[1] & Marcos M. Pires [1,2] ✉

The ability of biologically active molecules to access intracellular targets remains a critical barrier in drug development. While assays for measuring cellular uptake exist, they often fail to distinguish between membrane-associated or endosomal trapped compounds and those that successfully reach the cytosol. Here, we present the Chloroalkane HaloTag Azide-based Membrane Penetration (CHAMP) Assay, a high-throughput method that employs a minimally disruptive azide tag to report the cytosolic accumulation of diverse molecules in mammalian cells. The CHAMP assay utilizes HaloTag-expressing cells and strain-promoted azide-alkyne cycloaddition (SPAAC) chemistry to quantify the presence of azide-tagged test compounds in the cytosol. We demonstrate the versatility of this approach by evaluating the accumulation profiles of small molecules, peptides, and proteins, revealing how structural variations and stereochemical differences influence cytosolic penetration. Our findings with cell-penetrating peptides confirm established structure-activity relationships, with longer polyarginine sequences showing enhanced accumulation. Additionally, we observed that *C*-terminal amidation and D-amino acid substitutions significantly impact cellular penetration. When applied to supercharged proteins and antibiotics, CHAMP successfully discriminates between compounds with varying accumulation capabilities. This method provides a robust platform for screening cytosolic accumulation while minimizing the confounding effects of large tags on molecular permeability, potentially accelerating the development of therapeutics targeting intracellular pathways.

Small molecules, peptides, and proteins form the cornerstone of modern drug discovery, each offering distinct advantages in therapeutic applications. While small molecules constitute approximately 90% of pharmaceutical medications, proteins and peptides have emerged as promising alternatives due to their exceptional target specificity and favorable toxicity profiles[1]. Recent advances in peptide synthesis have significantly enhanced their stability and bioavailability, expanding their therapeutic potential[2–4]. However, a critical determinant of the therapeutic efficacy of any molecule is its ability to reach its intended target. For intracellular targets, the cell membrane represents a formidable barrier that molecules must overcome[5–7].

Developing effective therapies against intracellular targets presents significant challenges, particularly in quantifying molecular distribution within cells[8]. Lipinski's "rule of five" (Ro5) has traditionally guided predictions about passive diffusion into cells[8], yet small polar molecules and larger biomolecules (peptides, proteins, and nucleic acids) typically cannot readily traverse cell membranes directly[9]. Instead, these molecules primarily enter cells through endocytosis, often becoming trapped in endosomes rather than reaching their cytosolic targets[10]. The poorly understood process of endosomal escape further complicates therapeutic development, as conventional cell penetration assays struggle to differentiate between cellular uptake and cytosolic delivery. This highlights the urgent need for methods that can accurately measure cytosolic accessibility of potential therapeutics[10].

Current approaches to assess membrane permeability include direct quantification via LC-MS/MS or UV spectrophotometry[11,12], as well as standardized permeability assays such as PAMPA and Caco-2[13]. The PAMPA technique evaluates compound permeability through an artificial

[1]Department of Chemistry University of Virginia, Charlottesville, VA, USA. [2]Department of Microbiology, Immunology, and Cancer, University of Virginia, Charlottesville, VA, USA. ✉e-mail: mpires@virginia.edu

membrane separating donor and acceptor compartments[14], while the Caco-2 assay employs human colon adenocarcinoma cell monolayers to measure epithelial permeability[15,16]. Despite their utility, these methods have significant limitations: LC-MS/MS[17] offers excellent sensitivity but remains low-throughput[18] and cannot distinguish between membrane-bound compounds and those in the cytosol[19], while UV spectrophotometry lacks the sensitivity required for diverse molecular screening[20].

Fluorescent tagging represents an alternative strategy for visualizing molecular localization within cells[21]. These tags often substantially alter the physicochemical properties of the parent molecules, potentially modifying their biological activity, cellular localization, and dynamics[22,23]. More importantly, they cannot distinguish between whole cell association or cytosolic arrival. The Kodadek lab developed a high-throughput assay that used steroid fusions to evaluate the cell accumulation of synthetic molecules that produced a luciferase signal upon the ligand binding to ligand binding domain (LBD) of the glucocorticoid receptor[24,25]. More recently, the Kritzer laboratory developed the Chloroalkane Penetration Assay (CAPA), which utilizes a relatively smaller and more water soluble chloroalkane tag[26]. CAPA employs the genetically encoded HaloTag protein - a modified bacterial haloalkane dehalogenase that freely diffuses in the cytosol[27,28]. The assay involves treating HaloTag-expressing cells with chloroalkane-tagged test molecules, followed by a fluorophore-linked chloroalkane that undergoes a selective, rapid, and essentially irreversible reaction[27]. This approach generates reliable, standardized data with a direct correlation between fluorescence intensity and cytosolic accumulation, while its compatibility with flow cytometry enables high-throughput quantitative analysis[19]. More recently, the NanoClick assay was introduced. The NanoClick assay is a BRET-based assay used to quantify intracellular target engagement of small molecules. When the probe covalently binds the intracellular HaloTag-NanoLuc, the proximity of NanoLuc (donor) and the fluorescent reporter (acceptor) generates a BRET signal[29].

Despite widespread adoption of CAPA for measuring molecular accumulation in mammalian systems[30–32], a significant limitation persists: the long 15-atom chloroalkane chain tag may alter the permeability profiles and physicochemical properties of target molecules. To overcome this constraint, we have developed an improved approach that replaces the chloroalkane tag with a substantially smaller azide moiety[33], minimizing its influence on molecular permeability[33] (Fig. 1a). Our strategy demonstrates that proteins, peptides, and small molecules can be modified with an azide tag to effectively determine their cytosolic localization. Furthermore, we reveal how structural and stereochemical variations influence cytosolic accumulation in mammalian cells. Combined, we present a high-throughput, efficient method for accurately determining the cytosolic accessibility of molecules across wide ranges of chemical profiles (small molecules, peptides, and proteins).

## Results

Leveraging the subcellular localization of HaloTag in HeLa cells, we envisioned using the site-specific nature of this biochemical reaction to install a strained alkyne at defined cytosolic landmarks (Fig. 1b). For this step, cells are treated with the strained alkyne dibenzocyclooctyne (DBCO) linked to chloroalkane (DBCOcl). Therapeutics bearing azide groups that reach the cytosol could then be covalently captured by HaloTag through a strain-promoted azide–alkyne cycloaddition (SPAAC, Fig. 1c)[34,35]. Our laboratory previously demonstrated that we can utilize this combination of DBCO and azide to evaluate the arrival of molecules to the surface and to the peri-

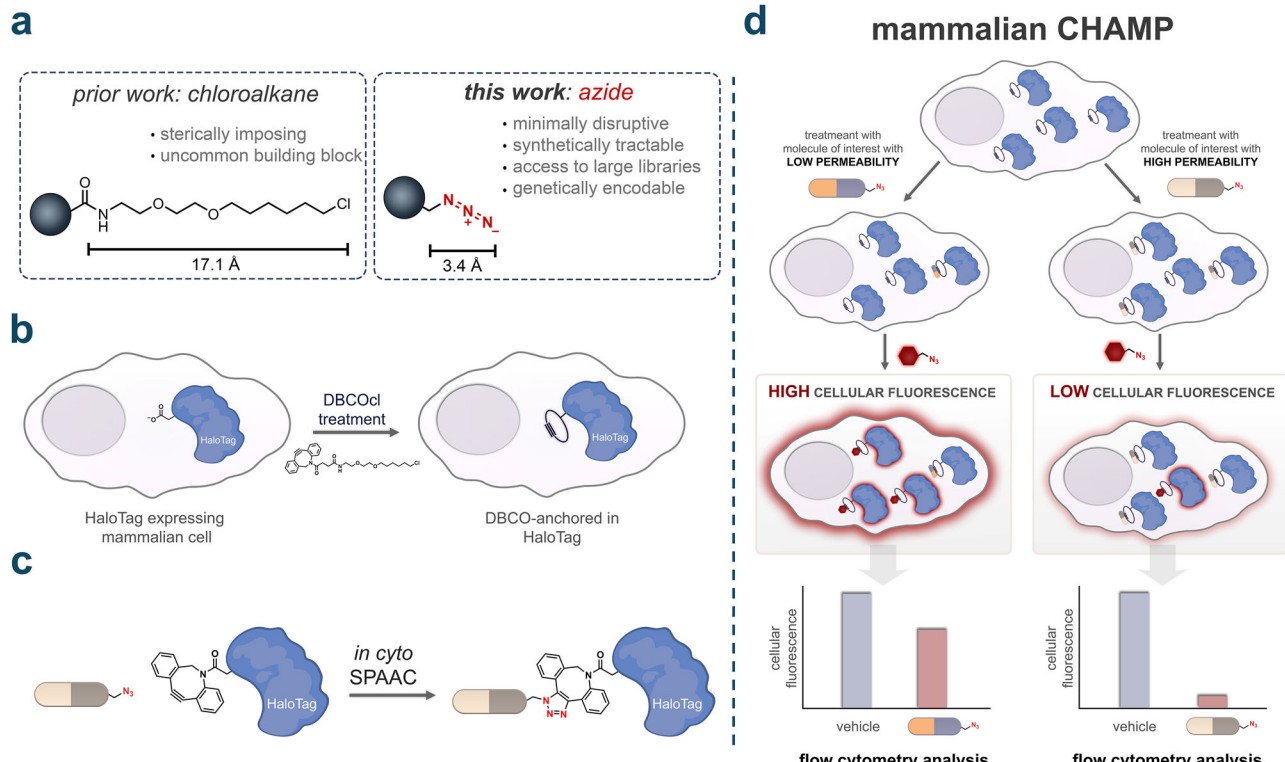

**Fig. 1 | Azide-based CHAMP strategy for quantifying cytosolic arrival in mammalian cells. a** Structural size comparison between two functional tags: Chloroalkane (HaloTag) and azide. **b** Representation of the installation of DBCOcl into HaloTag. **c** Arrival of compounds into the cytosol reacts with the strained alkyne handle via SPAAC reaction, demonstrating cytosolic localization. **d** Schematic representation of mammalian CHAMP workflow. First, cells expressing HaloTag in

the cytosol are treated with chloroalkane modified strained alkyne, followed by treatment with azide tagged molecule, "pulse step". Molecules with a high level of cytosolic arrivals leave with a lower number of DBCO active sites in exposure to fluorescent azide "chase step". Molecules with a high level of accumulation showed a lower level of cellular fluorescence.

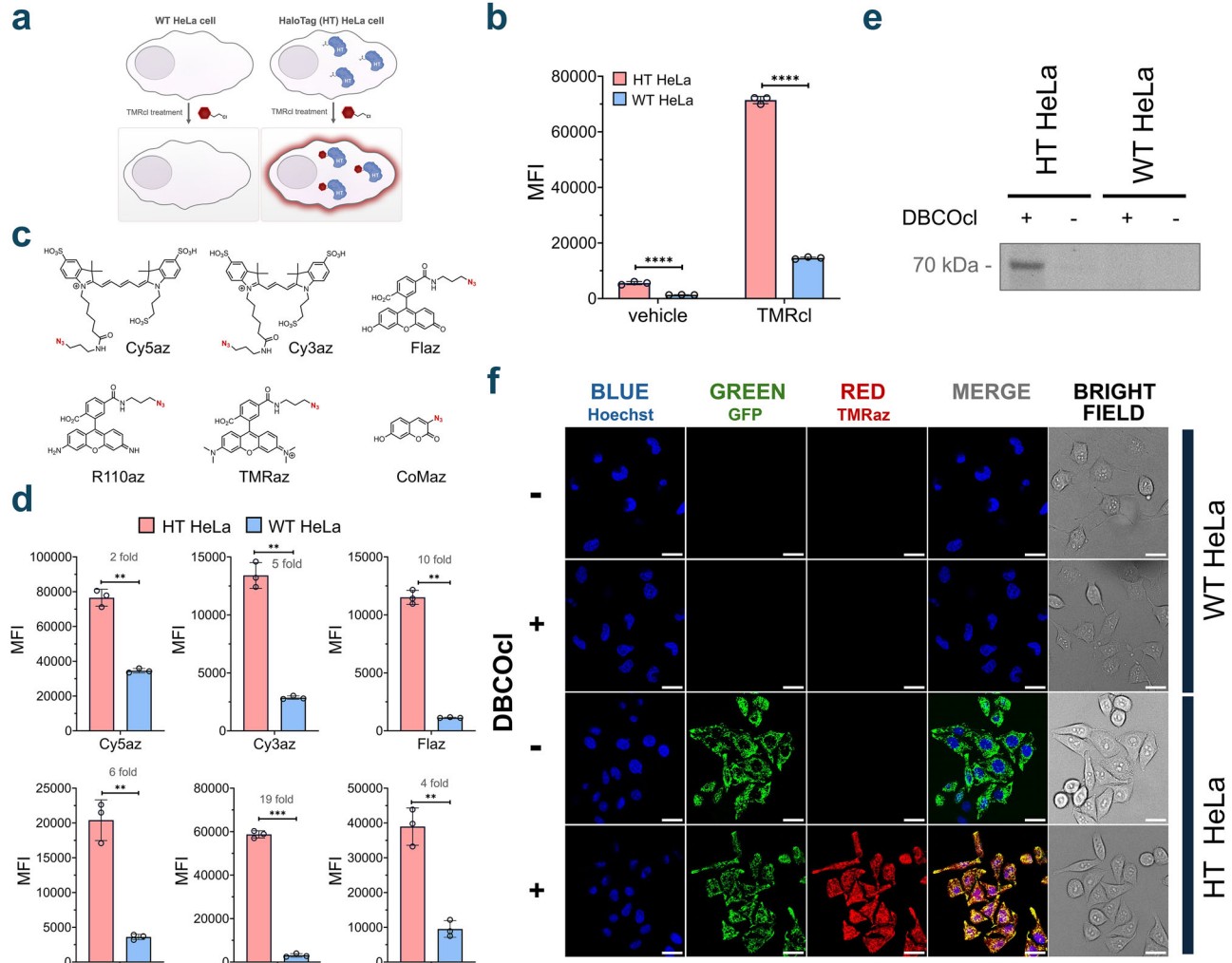

**Fig. 2 | Specific and tunable HaloTag–azide chemistry enables intracellular pulse–chase analysis. a** Schematic representation of the expression of HaloTag in HT HeLa cells. WT HeLa cells are used as a vehicle. **b** Flow cytometry data showing HaloTag expression. Cells were treated with 50 μM TMRcl and included no TMRcl control. WT HeLa cells were used as a negative control. Data are represented as mean ± SD (*n* = 3). **c** Chemical structures of azide-tagged fluorophores. **d** HT HeLa cells treated with six different azido-tagged fluorophores (TMRaz, COMaz, R110az, Flaz, Cy5az, and Cy3az). Cells were first treated with 10 μM DBCOcl for 15 minutes, washed and then treated with the 50 μM azido-tagged dyes for 15 minutes. WT HeLa cells were used as a negative control. Data are represented as mean ± SD (*n* = 3). **e** SDS-PAGE analysis of cells treated with and without DBCOcl followed by TMRaz. WT HeLa cells are used as a negative control. **f** Fluorescent microscopy images showing "pulse-chase" experiment with cells pulsed with DBCOcl and chased with TMRaz. WT HeLa and HT HeLa cells not treated with DBCOcl were used as negative control. The scale bar represents 25 μm.

plasmic space of bacteria[36,37]. Combined, these components result in the Chloroalkane HaloTag Azide-based Membrane Penetration (CHAMP) Assay (Fig. 1d)[38]. In this workflow, cytosolic accumulation of test molecules is evaluated using a pulse-chase format with azide-tagged fluorophores. After treatment, cells are analyzed by flow cytometry, and fluorescence intensity serves as a readout of cytosolic accumulation. Importantly, the fluorescence signal is inversely proportional to the permeability of the test molecule: highly permeable compounds react with and occupy most DBCO sites, resulting in low fluorescence, while poorly permeable compounds leave more DBCO landmarks available for reaction with the fluorophore, leading to higher fluorescence. This inverse relationship enables a quantitative assessment of the ability of a molecule to access the cytosol.

### Benchmarking parameters – CHAMP assay development

The presence of HaloTag in HeLa cells was first verified by treating cells stably transfected with a chloroalkane-tagged tetramethylrhodamine fluorophore (TMRcl), which covalently tags HaloTag. Wild type (WT) HeLa cells expressing only the green fluorescent protein (GFP) fusion served as negative controls (Fig. 2a). Flow cytometry analysis showed that there was

a significant increase in cellular fluorescence in cells expressing HaloTag upon the treatment with TMRcl (Fig. 2b). Confocal microscopy confirmed the localization of the HaloTag expression, which is fused to GFP, and it showed that there was considerable overlap between the GFP fluorescence and the signal from the chloroalkane-linked fluorophore (Supplementary Fig. 1). These observations demonstrated effective cytosolic retention of the chloroalkane-tagged fluorophore specifically in cells expressing HaloTag. We further characterized HaloTag expression through SDS-PAGE analysis of TMRcl-treated cells, which revealed a distinct fluorescent band at the expected molecular weight (~70 kDa) corresponding to the GFP-HaloTag fusion protein (Supplementary Fig. 2). We observe a single band that is fluorescent, which suggests that the covalent bond is almost exclusively at the site of HaloTag. Together, these results validated our cellular system for detecting chloroalkane-modified molecules in mammalian cells.

Then, the goal was to establish the installation of strained alkyne (DBCOcl) landmarks within HaloTag[39]. This installation can be revealed with an *in cyto* SPAAC reaction with an azide-tagged fluorophore. We evaluated six commercially available azido-fluorophores (TMRaz, COMaz, R110az, Flaz, Cy5az, and Cy3az) in both HaloTag-expressing and wild-type

HeLa cells (Fig. 2c) and monitored cell viability in the presence of each probe to ensure minimal cytotoxicity throughout the assay (Supplementary Fig. 3). Recognizing that their physicochemical properties could influence the extent of non-specific cellular binding, this feature was directly evaluated in HeLa cells. In each experiment, cells were first treated with DBCOcl to install DBCO onto HaloTag, followed by incubation with the respective azido-fluorophores. Among all candidates, TMRaz showed the highest fold increase in signal in the presence of HaloTag (Fig. 2d), making it our fluorophore of choice for subsequent experiments. We then examined whether the defined concentrations of TMRaz required for the assay compromised membrane integrity (Supplementary Fig. 4). Critically, these results suggest that unanchored DBCOcl in the cytosol does not significantly contribute to background signals. To confirm the covalent interaction between HaloTag-DBCO and TMRaz, we conducted fluorescent gel imaging on cells treated with or without DBCOcl followed by TMRaz (Fig. 2e; Supplementary Fig. 5). A fluorescent band appeared only when DBCOcl had reacted with HaloTag, confirming assay specificity. Confocal microscopy further validated the expected co-localization between the GFP signal from the HaloTag fusion and the clicked product of TMRaz (Fig. 2f). In HeLa cells expressing HaloTag, no apparent cellular fluorescence was observed in the absence of DBCOcl. However, when cells were treated with DBCOcl, a clear fluorescence signal emerged that overlapped with GFP signals. Collectively, these results indicate that the strained alkyne is primarily installed at HaloTag sites and that azide-tagged molecules react at the site of HaloTag.

We next aimed to determine the optimal DBCOcl concentration for efficient HaloTag labeling. Cells were treated with increasing concentrations of DBCOcl followed by a TMRaz chase step, which revealed that 10 µM of DBCOcl achieved apparent saturation of DBCO occupancy (Supplementary Fig. 6). In the absence of the DBCO anchor, fluorescence levels were near background; a finding that is consistent with minimal non-specific binding of the dye. To verify whether this concentration was necessary in the standard CAPA format, we performed a parallel DBCOcl concentration scan using chloroalkane-tagged TMR (TMRcl) in the chase step. Interestingly, a lower concentration of 4 µM DBCOcl generated a robust signal in this configuration (Supplementary Fig. 7). After establishing the working concentration parameters, we optimized incubation times. Cells were treated with DBCOcl at various times, and the cellular fluorescence was measured at each defined time point. Our results showed that the signal was effectively saturated by the first point of 2 minutes (Supplementary Fig. 8). These fast kinetics likely reflect a high level of accumulation efficiency, and the fast kinetics observed for HaloTag. To assess whether DBCOcl influences cellular permeability, a membrane integrity assay was performed. The result indicated no noticeable disruption of the plasma membrane even after 24 h of its incubation (Supplementary Fig. 9). Next, the incubation time and concentration for TMRaz was analyzed and our results showed that 50 µM (Supplementary Fig. 10) of TMRaz labeling was required to complete the reaction within 15 minutes of the incubation (Supplementary Fig. 11). These optimized parameters were then used to establish the saturation degree of HaloTag binding sites using flow cytometry and SDS-gel (Supplementary Fig. 12). Thus, these optimized parameters established working conditions for the CHAMP assay with maximum sensitivity and reproducibility.

**Screening of azide-modified small molecule compounds**

Having established the parameters for CHAMP, we next evaluated the accumulation profiles of a large untargeted library of azide-modified compounds in the mammalian cytosol. To accomplish this, we purchased a diverse library of azide fragments containing molecules in which the azide group resides in a wide range of electronic (aliphatic/aromatic) and steric (primary/secondary/tertiary) environments that can potentially modulate reactivity with alkynes. To account for this feature, we built a parallel platform whereby the strained alkyne landmark was covalently attached to a flow cytometry-compatible polystyrene bead (Fig. 3a). The landmarks on the beads were then subjected to pulse-chase steps that mirror the cell

treatments (same dye, same concentration, and same fluorophore). Therefore, relative changes in fluorescence signals (bead versus cells) should effectively isolate the contribution of the membrane bilayer in controlling the accessibility of the test compound to the strained alkyne (Fig. 3b). We measured in parallel both the bead fluorescence and cellular fluorescence of 384 compounds that varied in molecular weight, charge, polarity, hydrophobicity, rigidity, and number of hydrogen bond donors (HBD) and hydrogen bond acceptors (HBA) (Fig. 3c). For any given data point, the plot shows a specific molecule. When the whole-cell fold change is lower than the corresponding bead-based fold change, it suggests that the occupancy of the molecule on DBCO landmarks is impeded by the membrane of the cell. This comparison allows us to isolate and assess the impact of the membrane on each compound and its intrinsic reactivity with DBCO. This approach highlights the ability to adapt mammalian CHAMP to a multiwell plate format that can readily be employed in high-throughput workflows. While the generally small size of these fragment-like molecules prevented a deeper analysis that could broadly contribute to defining the accumulation determinants (e.g., in combination with machine learning algorithms), patterns of accumulation could be generally correlated with physicochemical parameters such as ClogP and TPSA (Topological Polar Surface Area) (Fig. 3d). Nonetheless, this level of throughput can be readily paired with a library that has the chemical space to more thoroughly probe how small molecule structure drives accumulation to the cytosol.

Next, we set out to probe how targeted structural edits, with a focus on charged states, can be systematically evaluated using CHAMP. The charged state of molecules can broadly impact desolvation profiles in the case of passive diffusion across the membrane bilayer. When charged species are important to impart biological activity, prodrug strategies are commonly employed to temporarily mask the charges, which can lead to enhanced drug delivery[40]. Given that carboxylic acids are amongst the most abundant functional groups found in pharmaceuticals, we benchmarked CHAMP by testing how the presence of an unprotected carboxylic acid affects the intracellular accumulation of small molecules in mammalian cells. To this end, we selected scaffold 1, comparing variants with either an amidated (1p) or free carboxylic acid (1n) at the C-terminus. As anticipated, the amidated derivative 1p exhibited greater accumulation than its free acid counterpart 1n (Fig. 4a). We then extended our analysis to a slightly larger scaffold (2) in which the amino group was positioned further from the C-terminus, thus avoiding potential internal hydrogen bonding configurations. A similar trend was observed: amidated 2p accumulated more efficiently than 2n, showing higher accumulation relative to scaffold 1, indicating smaller changes in structure may have significant effect on permeability (Fig. 4b). Finally, we investigated scaffold 3, which lacks an amino group, to assess whether the overall molecular charge – net neutral or negative in this case, as opposed to net positive or neutral in scaffolds 1 and 2 – influences cellular accumulation. As before, 3p outperformed 3n in accumulation (Fig. 4b). Comparable results were observed when the peptide series were evaluated at lower concentrations (Supplementary Fig. 13).

The methylation of amine groups can modulate the overall charge and hydrophobicity of a molecule, thereby influencing its cellular accumulation. The Hergenrother group has previously demonstrated that primary amines are privileged functional groups for promoting accumulation in E. coli[41]. To empirically assess how the degree of amine methylation affects accumulation in mammalian cells, we synthesized and evaluated scaffolds 4 and 5, each differing in amine methylation patterns. The 4 series was based on a scaffold similar to 3, with compounds 4b-4e containing progressively increasing methylation on the amine, and 4a bearing a terminal hydrocarbon in place of an amino group (Fig. 4c). Upon evaluation, compound 4a showed significantly lower accumulation compared to the un-, mono-, and di-methylated analogs 4b-4d, which exhibited comparable accumulation levels (Fig. 4d). In contrast, the trimethylated amine 4e displayed the poorest accumulation in the series. To validate these observations in a complementary assay, we synthesized series 5, which is structurally analogous to series 4, but functionalized with a small fluorophore (Fig. 4c). This modification enabled direct measurement of accumulation. Consistent with the

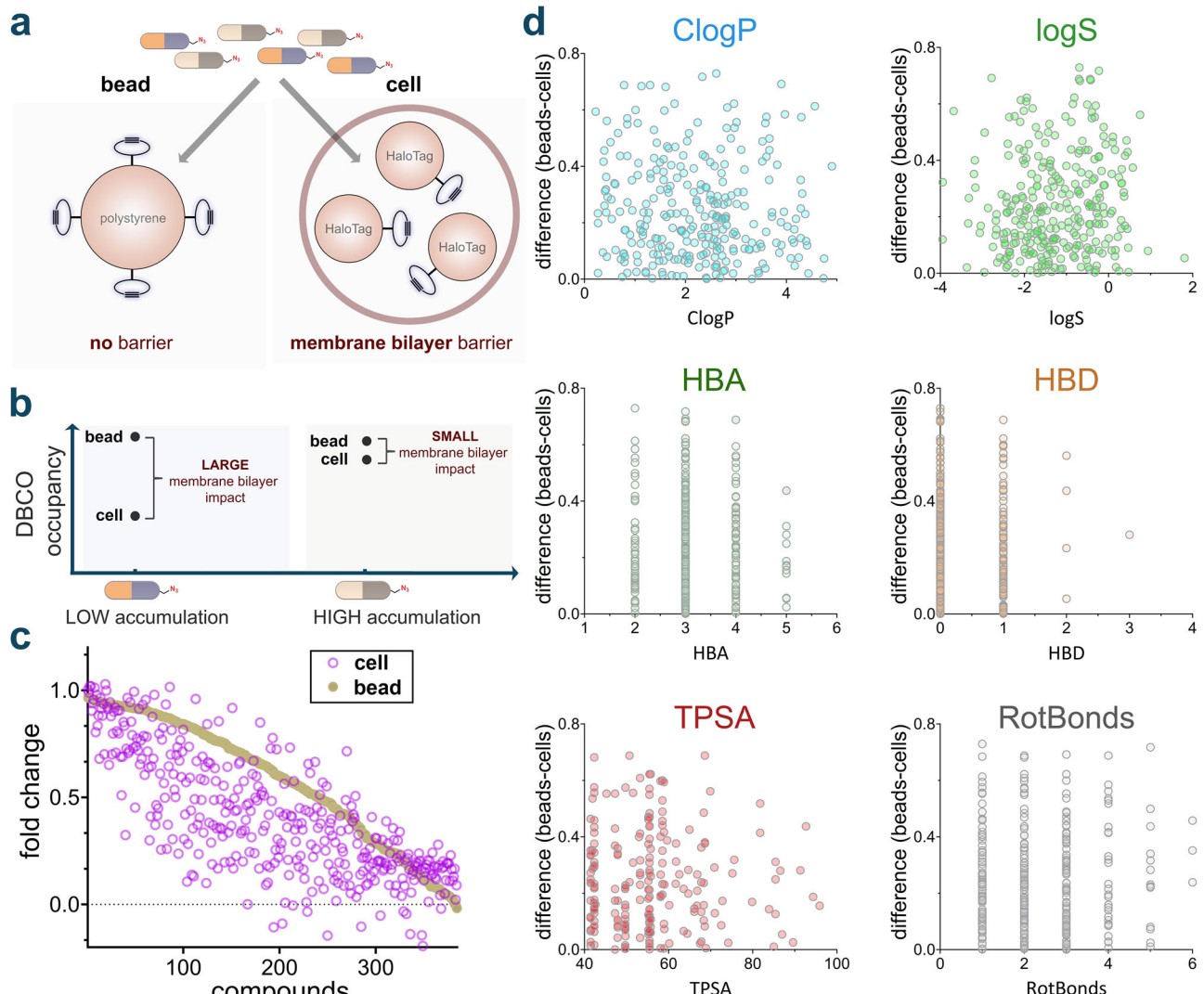

**Fig. 3 | Comparative analysis of intrinsic reactivity and cellular accumulation using CHAMP. a** Assessing probe permeability in beads (no barrier) versus live cells (membrane bilayer barrier). **b** Interpretation of membrane bilayer impact on DBCO labeling observed for both high permeable and low permeable compounds. **c** Comparison of apparent accumulation between bead and cell DBCO labeling of the 384-azido compound library. Accumulation relative to vehicle control is measured using the CHAMP assay, and reactivity relative to vehicle control is measured using the DBCO-tagged polystyrene beads assay. Data is represented as an individual data point ($n = 1$). **d** Effect of six different physicochemical parameters (ClogP, logS, HBA, HBD, TPSA, and RotBonds) on accumulation profiles of 384 azido library. For all CHAMP assays, HT HeLa cells were treated with 10 μM of DBCOcl, then incubated with 50 μM of the azide tagged molecules for 1 h, followed by treatment with the 50 μM TMRaz for 15 minutes. Data are represented as mean $+/-$ SD ($n = 3$). $P$-values were determined by a two-tailed $t$-test (* denotes a $p$-value $< 0.05$, ** $< 0.01$, ***$<0.001$, ****$<0.0001$, ns = not significant).

trends observed in series **4**, compound **5a** accumulated less than **5b**-**5d**, which showed comparable accumulation levels, while the trimethylated analog **5e** again exhibited the lowest accumulation (Fig. **4e**).

Several intracellular bacteria, including *Legionella pneumophila*, *Salmonella enterica*, and members of the *Mycobacterium* genus, reside and replicate in the cytosol of host cells, causing various bacterial infections[42]. We previously showed that antibiotics have a range of accumulation to the surface of bacteria once *S. aureus* is housed inside phagosomes, but that analysis was devoid of a cytosolic description. This is important because a subset of bacterial pathogens also resides in the cytosol. Antibiotic treatment of these intracellular bacteria often shows limited effectiveness. However, it remains unclear whether the failure of these well-known antibiotics is due to their inability to permeate the cell membrane and reach their targets or because bacteria become resistant to the antibiotics in this environment[43]. To evaluate the accumulation profiles of different antibiotics, we modified a panel of antibiotics with azide groups to make them compatible with the CHAMP workflow (Fig. **4f**). These antibiotics target various cellular

proteins, many of which are located in the cytosol[44]. For these antibiotics to be effective, they must cross the membrane barrier and reach their targets. Among the antibiotics with similar reactivity, significantly higher fluorescence signals were observed for all other than purAZ1, zolAZ3, and sulAZ1, suggesting increased cytosolic accumulation (Fig. **4g**).

**Accumulation studies of super-charged peptides**

We next utilized the CHAMP workflow to assess the cytosolic accumulation of cell-penetrating peptides (CPPs). Since the development of the first CPPs, Tat and penetratin, the field has rapidly advanced and continues to utilize the unique properties of CPPs to promote the delivery of cargo inside the cells[45–47]. CPPs are generally cationic or amphipathic peptides composed of relatively few residues (fewer than 30 amino acids) that efficiently transport a range of macromolecules across cell membranes when conjugated to them[48,49]. To evaluate the ability of CHAMP to report CPP accumulation in the cytosol of mammalian cells, we synthesized a series of azide-modified polyarginine peptides (R5az, R7az, R9az, and R11az, Supplementary

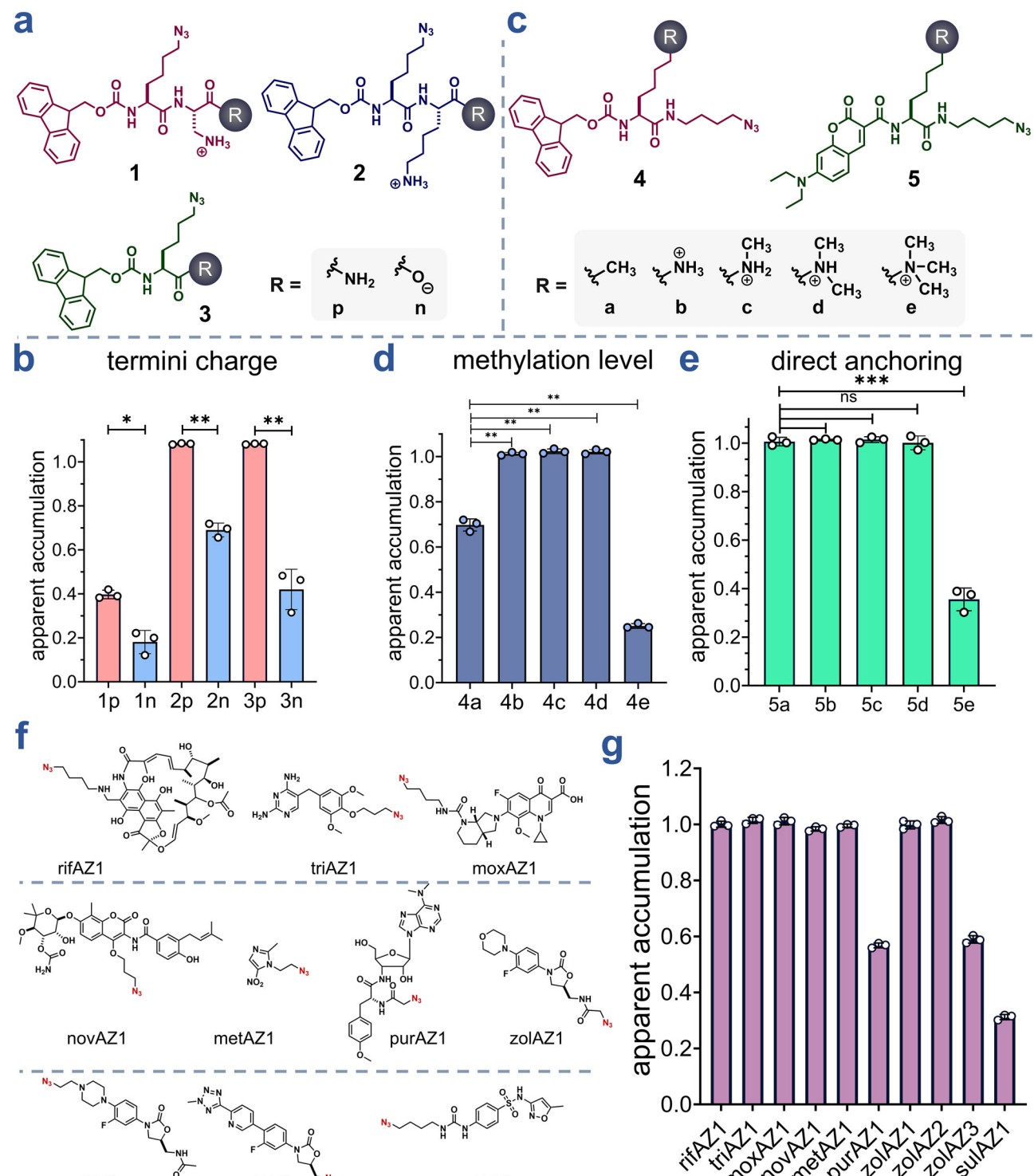

**Fig. 4 | Systematic modulation of charge, methylation, and anchoring strategy reveals determinants of cytosolic accumulation by CHAMP. a** Chemical structures of the 1–3 series. **b** Apparent accumulation of 50 μM of 1p, 1n, 2p, 2n, 3p, and 3n incubated for 1 h measured using the CHAMP assay. Data are represented as mean ± SD (n = 3). **c** Chemical structure of the 4–5 series. **d** Apparent accumulation of 50 μM of 4a-4d incubated for 1 h measured using the CHAMP assay. Data are represented as mean ± SD (n = 3). **e** Apparent accumulation of 50 μM of 5a-5d

incubated for 1 h measured using the CHAMP assay. Data are represented as mean ± SD (n = 3). **f** Chemical structure of azide modified antibiotics. **g** Comparative accumulation of 50 μM of azide-modified antibiotics incubated for 1 h measured using the CHAMP assay. Apparent accumulation was measured using the CHAMP assay. Data are represented as mean ± SD (n = 3). P-values were determined by a two-tailed t-test (* denotes a p-value < 0.05, ** < 0.01, ***<0.001, ****<0.0001, ns = not significant).

Fig. 14). Cells treated with these peptides at varying time points were tested to empirically determine the optimal experimental time course. Our findings revealed that a 24 h peptide incubation was required to achieve full apparent accumulation. However, prolonged incubation of CPPs might lead

to cytotoxic effects, so, 4 h incubation period was selected for further studies (Fig. 5a). We then performed a concentration scan of the peptides using this optimized time condition (Fig. 5b). The results demonstrated a positive correlation between the number of arginine residues (ranging from 5 to 11)

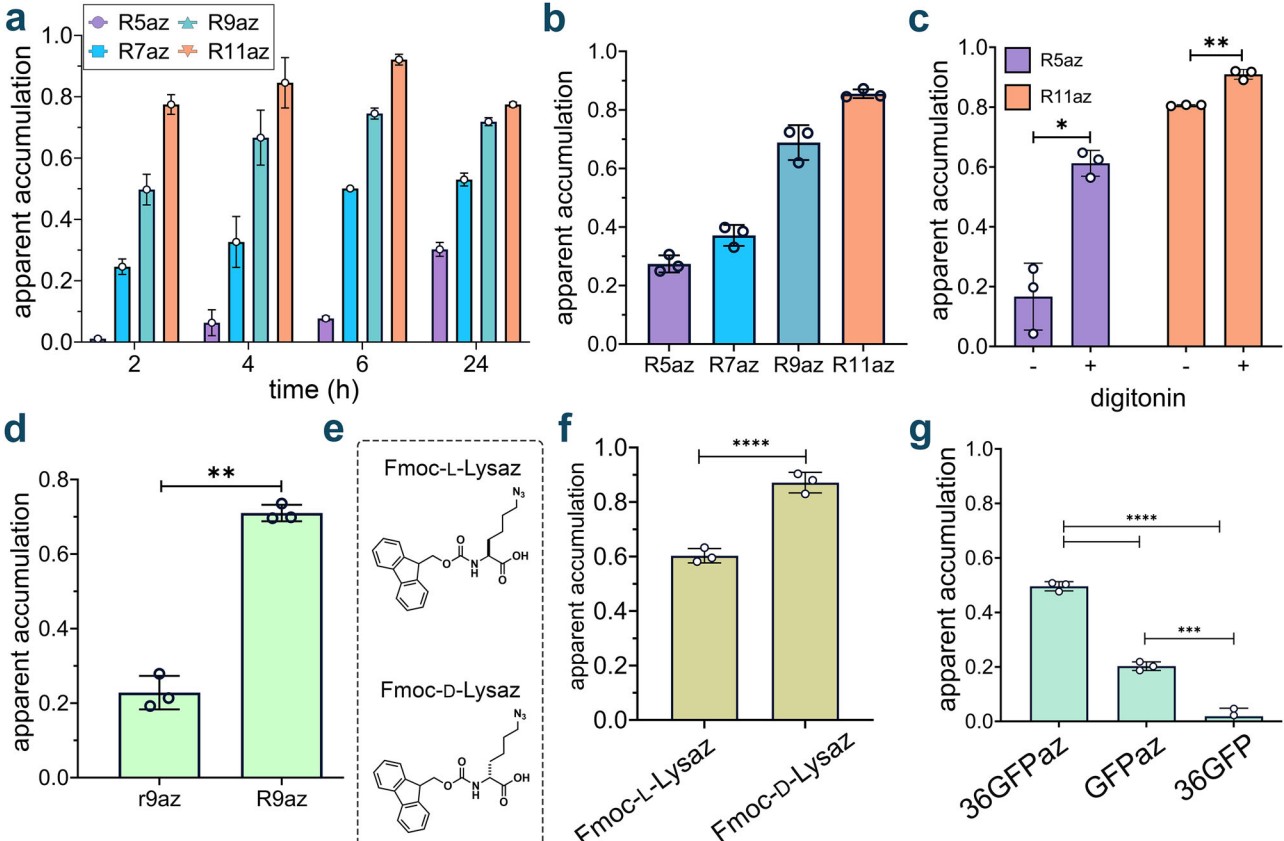

**Fig. 5 | CHAMP enables quantitative analysis of peptide and protein cytosolic accumulation. a** Optimization of the time of incubation of polyarginine peptides. Four different time points were tested: 2,4,6, and 24 h. Cells treated with 10 μM DBCOcl were pulsed with 50 μM of peptides and chased with 50 μM TMRaz. Data are represented as mean ± SD (*n* = 3). **b** Concentration scan of polyarginine peptides at six different concentrations: 2.5, 5, 10, 15, 25, and 50 μM. DBCOcl-treated HT HeLa cells were pulsed with 25 μM of peptides for 4 h and chased with 50 μM of TMRaz for 15 minutes. Data are represented as mean ± SD (*n* = 3). **c** CHAMP assay with and without 40 μg/mL of digitonin. DBCOcl-treated cells were pulsed with two peptides at 25 μM for 4 h with different accumulation profiles in the presence and absence of digitonin, followed by a chase step with TMRaz. Data are represented as mean ± SD (*n* = 3). **d** Comparison of the effect of stereochemistry on the accumulation of polyarginine peptides using the CHAMP assay. HT HeLa cells treated with

DBCOcl were pulsed with 25 μM of compounds for 4 h and chased with 50 μM of TMRaz. A higher fold change is indicative of higher relative accumulation. Data are represented as mean ± SD (*n* = 3). Data are represented as mean ± SD (*n* = 3). **e** Chemical structure of L and D Fmoc-Lysaz. **f** Comparison of the effect of stereochemistry on the accumulation of dipeptides using CHAMP assay. HT HeLa cells treated with DBCOcl were pulsed with 25 μM of compounds for 1 h and chased with 50 μM of TMRaz. Higher fold change is indicative of higher relative accumulation. Data are represented as mean ± SD (*n* = 3). **g** Apparent accumulation of 36GFPaz using CHAMP assay. Cells were treated with DBCOcl followed by treatment with 5 μM of supercharged protein for 6 h and chased with TMRaz. Unlabeled 36GFP was used as control. Data are represented as mean ± SD (*n* = 3). *P*-values were determined by a two-tailed *t*-test (* denotes a *p*-value < 0.05, ** < 0.01, ***<0.001, ****<0.0001, ns = not significant).

and cellular accumulation, consistent with previous reports[50–52]. Mechanistically, guanidine groups are believed to play a critical role in cell penetration by electrostatically interacting with sulfate, phosphate, and carboxylate moieties on the cell surface[53]. Our results indicated that R5az exhibited the lowest accumulation among the tested polyarginines, while both R9az and R11az showed the highest apparent accumulation. To understand the viability of cells in treatment with these peptides over optimized concentration and incubation time, we also performed the cytotoxicity assay of these peptides using the MTT (3-(4,5-dimethylthiazol-2-yl)-2,5-diphenyltetrazolium bromide) assay (Supplementary Fig. 15).

To further test the role of the membrane in reducing apparent accumulation, cells were treated with digitonin, a non-ionic detergent that permeabilizes membranes by complexing with membrane cholesterol[54], in the presence of polyarginine peptides. Before testing the peptides with digitonin, we first evaluated an azide-tagged dye with inherently low cellular permeability to confirm that membrane disruption facilitates intracellular entry. Based on prior reports, we selected sCy5az, a sulfonated cyanine molecule with an azide group, which is expected to exhibit minimal permeability due to its sulfonate groups[55]. Compared to cell treatment with dye alone, pre-incubation with digitonin significantly increased fluorescence signal (Supplementary Fig. 16). These findings highlight that the plasma membrane acts as a barrier to molecular entry, and its permeabilization enhances intracellular access. Further, they suggest that CHAMP can be used to monitor the integrity of the plasma membrane in the presence of potentially membrane-disrupting agents when paired with highly anionic dyes that are azide-tagged. Building on this result, we tested digitonin with polyarginine peptides, specifically R5az (low apparent accumulation) and R11az (high apparent accumulation). Our data showed that in the presence of digitonin, R5az displayed increased in apparent accumulation profile in the cytosol of mammalian cells (Fig. 5c). This indicates that digitonin successfully permeabilized the membrane, allowing the peptide to reach its target in the cytosol.

To better understand the dynamics underlying the CHAMP assay, we synthesized a CPP-based probe, CPPFlaz (R9LysazCoM, Supplementary Fig. 17). In this configuration, the peptide was labeled with both an azide tag and a fluorophore. The goal was to provide two independent measures of accumulation: one using the conventional direct-fluorophore readout, and the other using CHAMP analysis. The direct fluorescence measurement (associated with the coumarin dye) of CPPFlaz showed a clear dependence on both concentration and incubation time, reflecting progressive cytosolic accumulation (Supplementary Fig. 18). When the same cells were analyzed using CHAMP with the TMRaz for the pulse step, there was an increase in

apparent accumulation that was consistent with the coumarin-based signal. However, at higher peptide concentrations and with increased incubation time, only a modest change in TMR fluorescence was observed, whereas a pronounced increase in peptide fluorescence was detected. This discrepancy is likely attributable to signal contributions arising from nonspecific peptide binding or the possibility that the DBCO landmarks have been exhaustively occupied.

Additionally, we investigated the impact of stereochemistry on cytosolic accumulation in mammalian cells. Some models of permeation focus on the octanol–water partition coefficient[56], which can provide useful insights into the molecular state after complete solvation by the bulk solvent. Yet, these models fail to capture the potential interaction at the water-lipid interface that can operate with the stereochemical elements of phospholipids. This is important because, in theory, this interaction prior to permeation could impact the residency time and promote entry into the lipid bilayer. To this end, diastereoselective permeation was previously demonstrated in ribose using phospholipid bilayers, but not enantiomers of xylose[57]. We reasoned that we could leverage CHAMP to empirically test how the stereochemistry of molecules can alter cytosolic accumulation of molecules in live cells. Given that polyarginines are generally purported to interact with the headgroups of lipids in mammalian bilayers, we explored the role of stereochemistry in polyarginine accumulation by synthesizing all D-versions of R9, r9. Our data showed that R9 accumulated more efficiently in the cytosol of mammalian cells in both minimal media (Fig. 5d) and complete media (Supplementary Fig. 19). The use of complete media was to reduce the potential for protease-based degradation of the peptide during the incubation period. These results agree with previous results but additionally showed that the prior results also reflect the accumulation into the cytosol (as opposed to whole cell)[58,59]. Additionally, upon incubation with both R9az and r9az, more than 80% of the cells remained viable, indicating minimal cytotoxic effects (Supplementary Fig. 20).

Next, we took a complementary approach that focused on a non-peptidic small molecule. The goal was to evaluate how stereo configuration could potentially alter accumulation in live cells. For this series, we used Fmoc-L-Lysaz acid and Fmoc-D-Lysaz acid for their apparent accumulation to the cytosol of mammalian cells (Fig. 5e). Interestingly, we found that D-enantiomer had a higher level of apparent accumulation than its L-form counterpart, which may indicate stereospecific engagement with the headgroup of a lipid in a live cell (Fig. 5f). More recently[60], enantioselective accumulation of amino acids, also using click-chemistry to measure accumulation, was demonstrated in chiral lipid bilayers (not in a live cell), a finding that could be matched upon inversion of the stereocenters and lost upon disruption to the chirality of the lipid headgroups. Similarly, the enantiomeric counterpart of the antibiotic polymyxin was shown to be stereoselective in its engagement with the headgroup of lipid A on the surface of bacteria[61]. Together, our results highlight how CHAMP can be leveraged to interrogate features related to the three-dimensionality of a compound and its impact on accumulation across membrane bilayers.

There is growing recognition of the importance of developing protein-sized drugs, such as antibodies, nanobodies, and engineered protein scaffolds, that target intracellular pathways[62–64]. Traditionally, these larger biomolecules were confined to extracellular targets due to their limited ability to cross cellular membranes. We then postulated that our technique could be applied to analyze the accumulation profiles of larger macromolecules, such as supercharged proteins. Superpositively charged proteins, like +36 GFP, have been shown in multiple studies to accumulate at significantly higher levels than cationic peptides or mildly cationic designed proteins[65,66]. These proteins are known for their remarkable resistance to denaturants and their robust folding properties[67]. We anticipated that their positive charges would facilitate electrostatic interactions with the cell surface, thereby promoting internalization[66]. However, previous studies have shown that while supercharged proteins exhibit extensive endosomal uptake, only a small fraction successfully reaches the cytosol, and this efficiency varies depending on the cell line[68]. Critically, the installation of azides can be readily performed via metabolic techniques such as the incorporation of non-canonical amino

acids (ncAAs) by genetic dose expansion[69] or by Biorthogonal Non-Canonical Amino Acid Tagging (BONCAT)[70]. Alternatively, azide groups can be installed using electrophilic reagents such as N-hydroxysuccinimide (NHS).

To probe for cytosolic arrival of GFP variants, we expressed both wild-type and supercharged GFP (+36) and tagged these proteins with an azido tag using an azido-NHS ester. Gel analysis showed that both proteins and, upon reacting with TAMRA-tagged azide, fluorescence gel analysis showed that they were approximately the same level of labeling. Similar to the whole cell association results using confocal microscopy, we observed that positively supercharging the protein can promote their cytosolic arrival relative to wildtype GFP (Fig. 5g). Notably, the unlabeled +36GFP without the azide tag showed no accumulation, indicating that the azide tag was essential for the change in cellular fluorescence. Importantly, the total GFP fluorescence in cells remained comparable between labeled and unlabeled supercharged proteins, indicating that the azide modification does not alter overall GFP uptake or expression levels (Supplementary Fig. 21). Thus, our findings demonstrate that supercharged proteins can accumulate in the cytosol of mammalian cells, further validating the application of our technique for studying the intracellular delivery of macromolecules.

## Effect of macrocyclization and N-alkylation on peptide accumulation

Macrocyclization and N-alkylation are two commonly employed strategies utilized to enhance membrane permeability[71–75]. N-alkylation reduces the number of solvent-accessible hydrogen bond donors, thereby lowering the desolvation penalty associated with passive diffusion[76]. Similarly, macrocyclization can promote intramolecular hydrogen bonding, which further mitigates desolvation costs by shielding polar groups from solvent[73,77,78]. To systematically evaluate the impact of these modifications on peptide accumulation in mammalian cells, we tested a systematically varied library of azide-tagged macrocyclic and N-alkylated peptides.

First, we evaluated a sub-library of peptides of increasing sizes Cyc0–3, alongside their linear analogs Lin0–3 to gauge the influence of molecule size on accumulation (Fig. 6a)[79,80]. In all four pairs, the cyclic peptides generally exhibited greater intracellular accumulation compared to their linear counterparts. Additionally, both series displayed a size-dependent decrease in accumulation, with larger peptides accumulating to a lesser extent (Fig. 6b).

Peptides that are characterized by a looped or "lasso" structure are known as lariats and have recently been explored as drug scaffolds[81,82]. We evaluated a series of lariat peptides with decreasing ring sizes Lar1–5, along with a linear control Lar0 (Fig. 6c). Consistent with previous observations, the cyclic peptides generally outperformed the linear analog. Although no clear trend emerged across the cyclic series, the peptide with the smallest ring – Lar5, exhibited the highest accumulation (Fig. 6d). This result was unexpected, as smaller rings typically expose more surface area to solvent, potentially diminishing the accumulation advantages conferred by cyclization.

To evaluate whether an alternative cyclization strategy could also enhance accumulation, we assessed the Dit0–2 series (Fig. 6e). Dit1, a disulfide-containing peptide, was generated by oxidizing a dithiol scaffold. Using the same linear precursor, we synthesized Dit2 via cyclization with a bis-electrophilic linker, yielding two thioether bonds. Dit0 served as the linear control. As observed previously, both cyclic peptides accumulated more effectively than the linear analog (Fig. 6f). Notably, Dit1 outperformed Dit2, underscoring the impact of cyclization chemistry on accumulation. This difference may arise from variations in conformation, structural rigidity, or hydrophobicity-factors that can influence membrane permeability[83].

Within the N-alkylated library, we first examined how the degree of N-methylation affects peptide accumulation by evaluating peptides Nmet1–5 in comparison to their unmethylated counterpart, Nmet0 (Fig. 7a). A modest increase in accumulation was observed with the addition of up to two N-methyl groups (Fig. 7b). Beyond this point, no significant differences

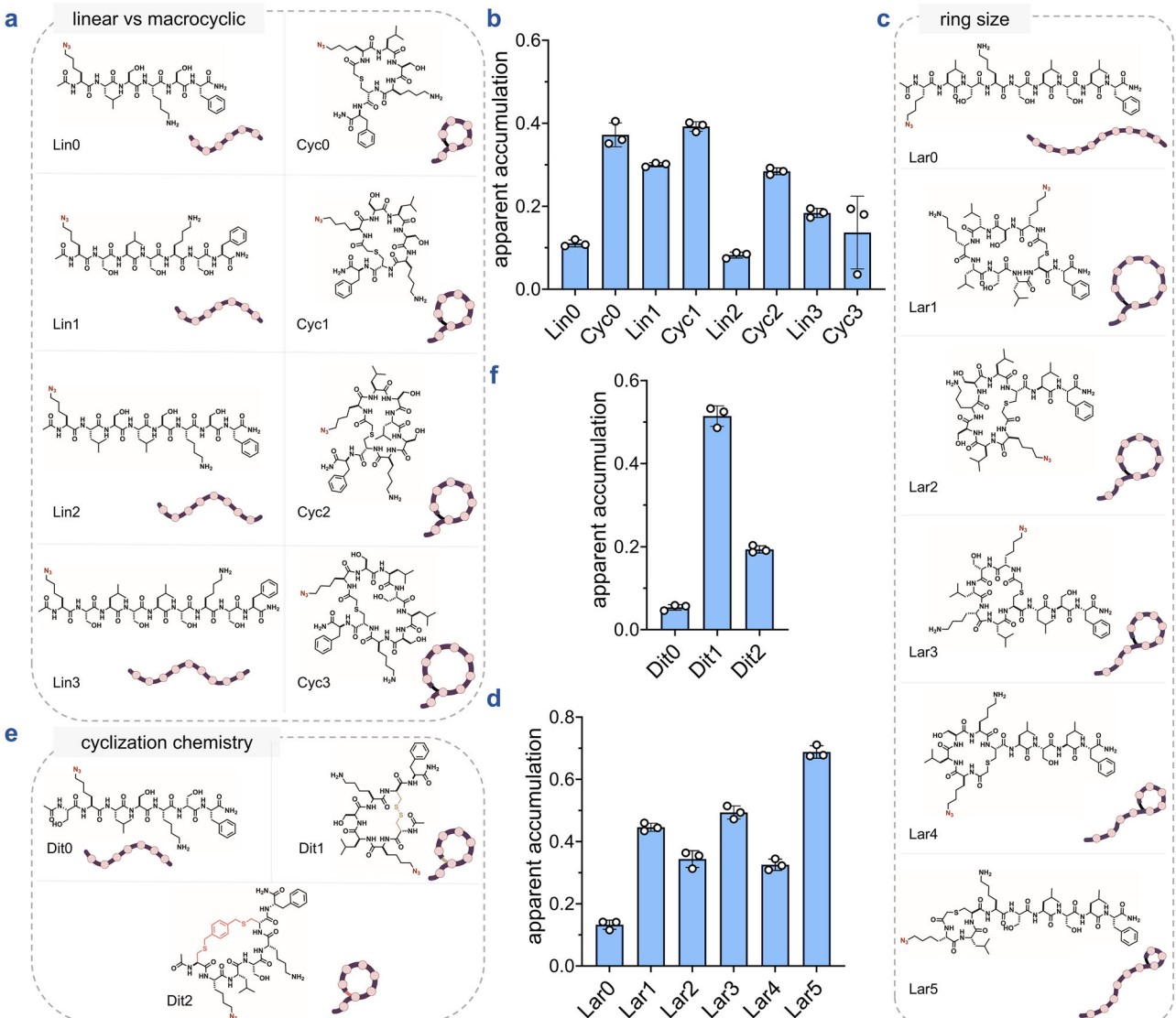

**Fig. 6 | Macrocyclization, ring size, and cyclization chemistry modulate cytosolic accumulation of peptide libraries measured by CHAMP. a** Chemical structures of the linear vs macrocyclic sub-library members containing macrocyclic peptides of increasing sizes (Cyc0-3) and linear counterparts (Lin0-3). **b** Apparent accumulation of linear vs macrocyclic sub-library using CHAMP assay. HT HeLa cells treated with 10 μM of DBCOcl were pulsed with 50 μM of compounds for 24 h and chased with 50 μM of TMRaz. Data are represented as mean ± SD ($n = 3$) **c** Chemical structure of the ring size (lariat) sub-library members containing macrocyclic peptides of increasing ring sizes (Lar1-5) and their linear counterpart (Lar0). **d** Apparent accumulation of the ring size sub-library. HT HeLa cells treated with

DBCOcl were pulsed with 50 μM of compounds for 24 h and chased with 50 μM of TMRaz. Data are represented as mean ± SD ($n = 3$). **e** Chemical structures of the cyclization chemistry sub-library members containing a disulfide-bonded macrocyclic peptide (Dit1), a bis-electrophilic linker-based macrocyclic peptide (Dit2), and their linear counterpart (Dit0). **f** Apparent accumulation of the cyclization chemistry sub-library. HT HeLa cells treated with DBCOcl were pulsed with 50 μM of compounds for 24 h and chased with 50 μM of TMRaz. Data are represented as mean ± SD ($n = 3$). *P*-values were determined by a two-tailed *t*-test (* denotes a *p*-value < 0.05, ** < 0.01, ***<0.001, ****<0.0001, ns = not significant).

were noted among peptides containing two, three, or four *N*-methyl groups. However, a slight decrease in accumulation was observed when five *N*-methyl groups were introduced. Upon evaluating peptides bearing a single *N*-methylation at different backbone amide positions (**Nmet6-Nmet9** and **Nmet1**) (Fig. 7c), we observed clear positional effects on cytosolic accumulation, with **Nmet9** showing the highest level of uptake (Fig. 7d). Importantly, positional and degree effects of *N*-methylation have previously been observed[84–89] underscoring the importance of empirical evaluation when optimizing *N*-methylation patterns for intracellular delivery.

Peptoids, which are *N*-substituted glycine oligomers bearing side chains on the backbone nitrogen rather than the α-carbon, have been investigated as scaffolds for targeting drug-resistant bacterial pathogens[90–93]. In this context, we tested a series of peptoids analogous to our *N*-methylated peptide library (Supplementary Fig. 22). As observed previously,

accumulation increased slightly with the addition of up to two methylation marks, with no further improvement beyond that point. Notably, in assessing positional effects, **Nalk9** analogous to the top-performing positional series *N*-methylated peptide, **Nmet9** emerged as the most privileged scaffold in the series (Supplementary Fig. 23). Overall, our analysis revealed that macrocyclization generally enhances accumulation into the mammalian cytosol, whereas the effects of *N*-methylation are more context- and position-dependent, varying across different substitution patterns.

## Conclusions

In conclusion, the CHAMP assay builds upon the foundational CAPA methodology to evaluate the intracellular accumulation of azide-tagged molecules, offering a minimally disruptive alternative to chloroalkane tags, which could otherwise affect uptake and intracellular distribution. CHAMP

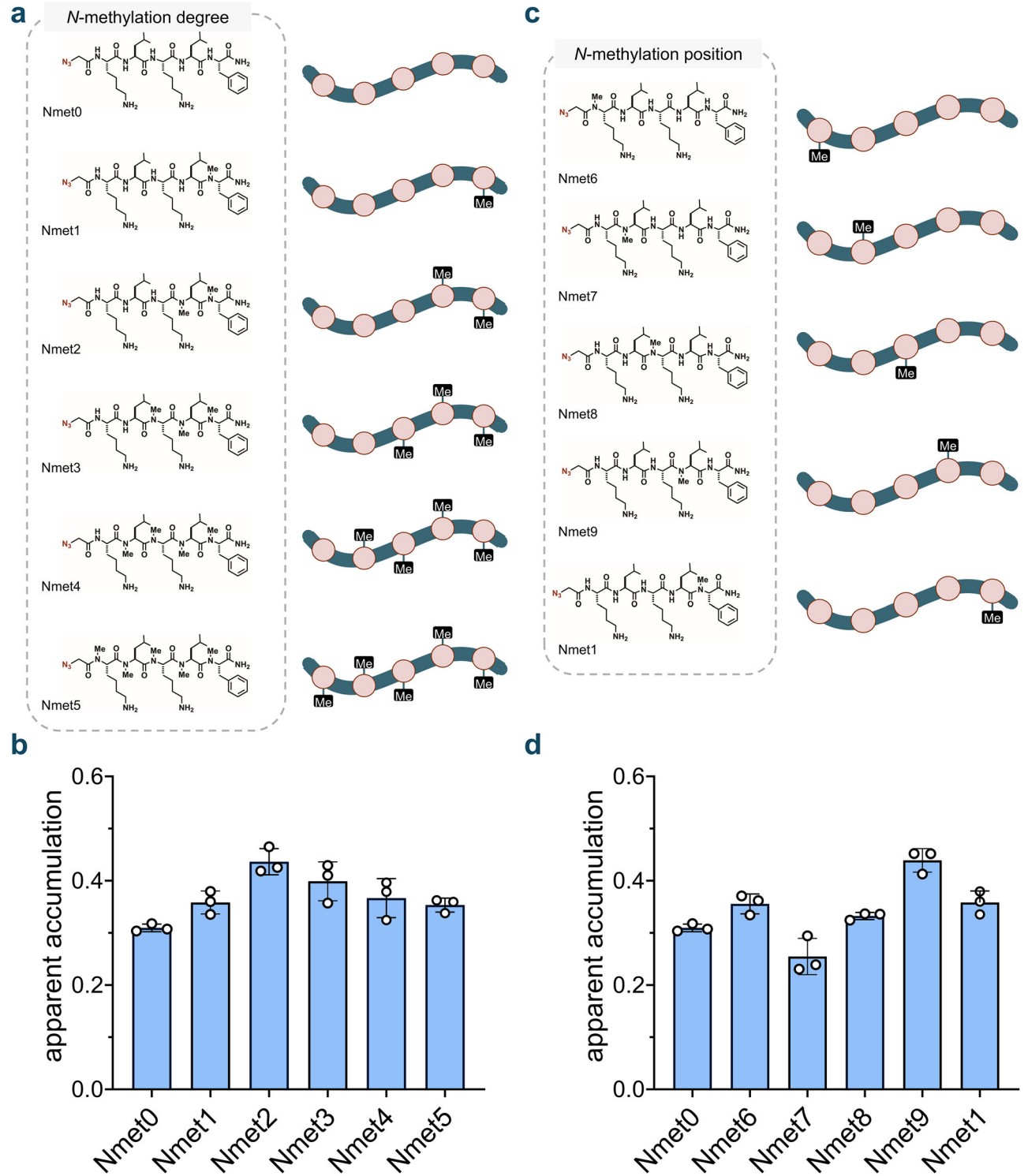

**Fig. 7 | Degree and positional effects of N-methylation on peptide cytosolic accumulation measured by CHAMP. a** Chemical structures of the *N*-methylation sub-library with varying degrees of backbone *N*-methylation (Nmet0-5). **b** Apparent accumulation of the *N*-methylation sub-library with varying degrees of backbone N-methylation using the CHAMP assay. HT HeLa cells treated with DBCOcl were pulsed with 50 µM of compounds for 24 h and chased with 50 µM of TMRaz. Data are represented as mean ± SD (*n* = 3). **c** Chemical structures of the *N*-methylation sub-library with varying positions of backbone *N*-methylation (Nmet6-9 and Nmet1). **d** Apparent accumulation of the *N*-methylation sub-library with varying positions of backbone *N*-methylation. HT HeLa cells treated with DBCOcl were pulsed with 50 µM of compounds for 24 h and chased with 50 µM of TMRaz. Data are represented as mean ± SD (*n* = 3). *P*-values were determined by a two-tailed *t*-test (* denotes a *p*-value < 0.05, ** < 0.01, ***<0.001, ****<0.0001, ns = not significant).

reliably quantifies the cytosolic penetration of a wide range of compounds, including small molecules, peptides, and proteins. Compared to the previously described Nanoclick method, CHAMP offers several key advantages through its flow cytometry-based readout. Flow cytometry provides single-cell resolution rather than bulk population averages, enabling identification of heterogeneous cellular responses and gating on relevant cell subsets. This approach also offers better control over expression variability, higher signal intensity and dynamic range, and the ability to multiplex with viability dyes

or phenotypic markers. These features yield cleaner, more interpretable data and enable linkage of functional readouts to specific cell populations, capabilities not readily achievable with plate-reader BRET measurements. Thus, the choice between methods may depend on the specific experimental requirements and available instrumentation. We showed that CHAMP effectively identifies variations in accumulation profiles based on structural and stereochemical differences, supporting comparative studies of azide-modified compounds in mammalian cells.

Furthermore, we envision CHAMP as a tool capable of reporting accumulation differences across various subcellular organelles using signal peptides. This would enable the measurement of molecular accumulation in the nucleus, outer mitochondrial membrane, endoplasmic reticulum, Golgi apparatus, peroxisomes, or lysosomes. Such capabilities could provide valuable insights into the role of mono- or bilayer subcellular membranes in molecular permeation, ultimately advancing the development of effective intracellular therapies by overcoming the challenges of measuring cytosolic accumulation with small tags. However, as with other cellular accumulation methods (including LC–MS/MS, NanoClick, and CAPA), molecules of interest may undergo degradation, resulting in an apparent total signal that differs from that of the intact parent compound. This limitation is inherent to these assay formats. Users should therefore consider assessing whether, and to what extent, degradation occurs in their system, as this may influence the interpretation of the biological readout.

## Methods

### Reagents
HeLa cells expressing a GFP-HaloTag fusion protein (HaloTag HeLa: HT HeLa) were generously provided by the Chenoweth Lab at the University of Pennsylvania. Dulbecco's Modified Eagle Medium (DMEM), Fetal Bovine Serum (FBS), penicillin-streptomycin, puromycin, and digitonin were obtained from Sigma-Aldrich. A library of 380 azide-containing small molecules was purchased from Enamine (catalog # AZD-380-X-100). Hoechst 33342 and HEPES were acquired from Thermo Fisher Scientific. The plasmid encoding pET-6xHis-GFP (+36) was obtained from Addgene (plasmid #199165). DBCONHS ester was purchased from Click Chemistry Tools.

### Cell culture
HT HeLa cells were cultured in Dulbecco's Modified Eagle Medium (DMEM) supplemented with 10% fetal bovine serum (FBS), 1% penicillin-streptomycin, and 1 µg/mL puromycin. Cells were maintained at 37°C with 5% $CO_2$. WT HeLa cells were cultured under identical conditions, except without puromycin selection. HaloTag HeLa cells were passaged using trypsin upon reaching 80–90% confluency and resuspended in DMEM. Cells were seeded in 96-well plates at a density of 50,000 cells per well, incubated for 24 hours at 37°C with 5% $CO_2$.

### One-step assay with dye (TMRcl/TMRaz)
HT HeLa cells were seeded into 96-well plates one day before the assay, reaching 80–90% confluency. Following overnight incubation at 37°C, cells were washed three times with PBS and treated with 10 µM DBCOcl in media for 15 minutes which is followed by chase with 50 µM of a dye in media (TMRcl/TMRaz) for 15 minutes. Then, the cells were washed three times using PBS, trypsinized for 5 min, and fixed with 4% formaldehyde in PBS for 30 min statically at room temperature. The cells were then subjected to analysis by flow cytometry.

### Chloroalkane HaloTag Azide-based Membrane Penetration (CHAMP) Assay
HT HeLa cells were seeded into 96-well plates one day before the assay, reaching 80–90% confluency. Following overnight incubation at 37°C, cells were washed three times with PBS and treated with 10 µM DBCOcl in media for 15 minutes. This was followed by a pulse with azide-tagged compounds diluted in media at required concentration, then a chase step with 50 µM TMRaz in media for 15 minutes. Cells were washed three times with PBS

between each step. After staining, cells were washed again, trypsinized, fixed with 4% formaldehyde, and analyzed using flow cytometry on the Attune NxT Acoustic Focusing Cytometer (Invitrogen).

### Confocal fluorescence microscopy
Confocal images were acquired using a Leica SP5X laser scanning microscope and a Leica STELLARIS 8 confocal/FLIM/tauSTED system equipped with tunable white light lasers and a 37-2 digital temperature controller. Image acquisition was performed using the LAS-AF software, and Fiji (ImageJ) was used for further processing. HeLa Halo and unmodified HeLa cells were trypsinized upon reaching 80–90% confluency, resuspended in DMEM, and seeded at a density of 50,000 cells on a 10 mm plate. After 24 hours of incubation at 37 °C with 5% $CO_2$, one-step assay with dye was performed, then cells were stained with Hoechst 33342 (Thermo Fisher, H3570) following the manufacturer's instructions. Cells were subsequently washed twice with PBS, rinsed three times with pre-warmed DPBS, and incubated in 1.5 mL DMEM containing 25 mM HEPES (Thermo Fisher, 21-063-029) for imaging.

### Screening of azide-modified compound library
The azide-modified compound library was screened at 50 µM for 1 hour using the CHAMP assay as described above. Fluorescence data were corrected for background contributions by subtracting signals from blank controls and normalized to the maximum fluorescence signal.

### Membrane permeabilization assay
Digitonin (Sigma, Part No. D141-100MG) was used for the membrane permeabilization assay. For this assay, 40 µg/mL digitonin was co-incubated with peptides for followinga 15-minute pulse with 10 µM DBCOcl. This was followed by a chase step with 50 µM TMRaz for 15 minutes. Cells were then trypsinized, fixed with 4% formaldehyde, and analyzed using flow cytometry.

### MTT cell viability assay
HT HeLa cells were seeded into 96-well plates one day prior to the assay to achieve 80–90% confluency. After overnight incubation at 37°C, cells were washed three times with PBS and treated with 25 µM polyarginine peptides for 4 hours at 37 °C. Subsequently, 0.45 mg/mL MTT solution in media was added, and cells were incubated for 2 hours at 37 °C. Following incubation, the plates were centrifuged, the supernatant was removed, and the resulting formazan crystals were dissolved in 100 µL of DMSO. Absorbance was measured at 570 nm using a plate reader.

### +36GFP amino acid sequence
MGHHHHHHGGASKGERLFRGKVPILVELKGDVNGHKFSVRGKGK
GDATRGKLTLKFICTTGKLPVPWPTLVTTLTYGVQCFSRYPKHMKR
HDFFKSAMPKGYVQERTISFKKDGKYKTRAEVKFEGRTLVNRIKLK
GRDFKEKGNILGHKLRYNFNSHKVYITADKRKNGIKAKFKIRHNVK
DGSVQLADHYQQNTPIGRGPVLLPRNHYLSTRSKLSKDPKEKRDH
MVLLEFVTAAGIKHGRDERYK.

### Expression and purification of +36GFP and wild-Type GFP
Plasmids encoding pET-6xHis-GFP (+36) (addgene #199165) were transformed into Escherichia coli BL21 cells and inoculated into 1 L of Lysogeny Broth (LB) supplemented with 100 µg/mL ampicillin. Cultures were grown at 37 °C with shaking at 250 rpm until reaching an OD600 of 0.5. Protein expression was induced by the addition of 1 mM IPTG, followed by incubation at 37 °C for 3 hours and then overnight at 24 °C. Cells were harvested by centrifugation at $4000 \times g$ for 15 minutes at 4°C. All subsequent purification steps were performed on ice. Cell pellets were lysed by sonication (Fisher Scientific FB505) for 25 minutes in cycles of 3 minutes on and 5 minutes off. Cell debris was removed by centrifugation at $4000 \times g$ for 20 minutes at 4 °C, followed by an additional centrifugation at $23,000 \times g$ for 20 minutes to separate soluble and insoluble fractions. Supercharged proteins were purified under native conditions using Ni-NTA agarose

chromatography. The wash buffer contained 50 mM phosphate buffer, 1 M NaCl, and 35 mM imidazole (pH 8.0), while the elution buffer contained 50 mM phosphate buffer, 1 M NaCl, and 250 mM imidazole (pH 8.0). Purified proteins were concentrated using an Ultra-15 centrifugal filter unit (Millipore, UFC901024) and dialyzed in 50 mM phosphate-buffered saline (pH 8.4). The azide tag was introduced using azidoacetic NHS ester (Click Chemistry Tools, Az103-100).

## Uncropped and unprocessed scans

Full scan gels are available on the Figshare public repository and can be downloaded with no restrictions at https://doi.org/10.6084/m9.figshare.30935729.

## Statistics and reproducibility

Experiments using fluorescence microscopy, SDS–PAGE, and flow cytometry were performed in a minimum of three independent biological replicates, each yielding comparable outcomes. For gel-based analyses, technical replicates from three parallel wells were combined prior to protein denaturation and loading for SDS–PAGE. Sample sizes were not determined using formal statistical power calculations but were selected in accordance with standard practices in chemical biology literature. Flow cytometry measurements were acquired with at least 10,000 events per sample, a widely accepted standard that ensures reliable estimation of population-level signals under common counting statistics assumptions.

## Reporting summary

Further information 641 on data collection and sample size is available in the 642 Reporting Summary linked to this article.

## Data availability

The data sets generated and/or analyzed during the current study are available within this article. The Supplementary Information files included as Supplementary Data 1, Supplementary Data 2, Supplementary Data 3, and can be downloaded with no restrictions at https://doi.org/10.6084/m9.figshare.30935729.

## Code availability

All raw data supporting the findings of this study are available within the article, and its Supplementary Information files included as Supplementary Data 1, Supplementary Data 2, and Supplementary Data 3. Flow cytometry data is available in a spreadsheet that can be downloaded at https://doi.org/10.6084/m9.figshare.30935729.

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

## Acknowledgements

This study was supported by the NIH grant 1R01AI178975-01 (M.M.P., W.I, and S.S.), R35GM124893 (M.M.P.), R01AI179080-01 (M.M.P., W.I, and S.S.), and the NSF grant MCB-2111728 (W.I.).

## Author contributions

S.B., G.M.O., R.D., Z.L., M.D.C., Y.H., and M.M.P. designed the research and contributed to experimental design. S.B. performed all primary mammalian experimental work. G.M.O. supported the initial mammalian work, including preliminary confocal and flow cytometry studies. R.D. and Z.L. contributed to the peptide libraries. M.D.C. contributed to the synthetic work of the small molecules. S.B., R.D., and M.M.P. contributed to the figures of the manuscript and the writing. All authors discussed the experimental results.

## Competing interests
The authors declare no competing interests.
