## [Transparent Peer Review file · Communications Chemistry]

A Generalizable assay for intracellular accumulation to profile cytosolic drug delivery in mammalian cells

Corresponding Author: Professor Marcos Pires

Version 0:

Reviewer comments:

Reviewer #1

(Remarks to the Author)

This is a paper that represents a potentially valuable improvement of the CAPA assay to measure cell penetration. Briefly, this assay involves an azide label on cell penetrating agents, as opposed to a chloro tag. The azide is smaller than the chlorotag. Conceptually, this likely means that the tag interferes less with the cell penetrating agents. This is likely for small penetrating agents, less likely for larger molecules. To increase potential impact, the authors should certainly test the effect of Azide vs Chlorotag on the cell penetration of a model compounds. For large molecules like proteins, an azide could be useful because it can be integrated with unnatural amino acid incorporation. One of the downside of the approach is the need for cell pretreatment with a intermediate step that prime the HaloTag reporter. The effect of this treatment on cell penetration is lacking in the study. This is critical as the claims on cell penetration depends on this step not modifying membranes at all.

Part of the paper is to establish the technology, part is to learn about cell penetration. The paper generally lacks technical details that would convincingly demonstrate that cell penetration does not result from artifacts (e.g. cell death, effect of DBCOcl treatment - see below).

Figure 1 and related text:

- The CHAMP assay is an excellent idea to minimize the size of the chloroalkane on probes with CAPA experiments; an azide instead of a chloroalkane definitely appears to be less problematic for the compound being studied
- There is no example of the raw data generated by the CHAMP assay in this figure, which can make it confusing to understand what is being measured. It would be helpful to show an example of the changing fluorescence of the azide chase from flow cytometry or SDS-PAGE with low or high permeability compounds. Also, how quantitative measurements of cell penetration are derived from flow cytometry should be presented, along with associated statistical analysis,
- Is it possible that DBCOcl alters the permeability of cells in the CHAMP assay at all? In the CAPA experiment, the chlorotag chase is done only after incubation with the compound being tested. However the CHAMP assay requires DBCOcl treatment prior to compound treatment. The potential influence of this pre-treatment should be tested. Different experiments use different wait times between DBCOcl and penetration assays. Hence, the potential effects of DPCOcl should be tested for all these conditions.
- The process of synthesis of compounds or peptides with an azide tag is not discussed at all; this detail is crucial for the CHAMP assay. Are azides easy or limited with attachment to compounds? If this process is easy, this should be explained as a strength of the assay. If it is a limitation, that should also be explained.
- S17 (Methods):
 - o What buffer are the DBCOcl, azide-tagged compounds, and TMRaz in during incubation with cells? Is PBS being used? Or perhaps L-15 buffer? This should be noted
 - o Are dead cells removed from analysis at all? Could it be that dead cells lead to a misleading increase in permeability?

Figure 2 and related text:

- Testing the different compounds for the az chase is an essential step for the CHAMP assay
- Background binding of the az chase to cells (non-HaloTag cells) appears very low
- Could this CHAMP approach be validated with a regular CAPA experiment? This could be one way to show that the CHAMP assay is consistent with previous strategies and that the DBCOcl treatment is not altering the results
- More raw output data from the chase should be shown (either from flow cytometry or SDS-PAGE) to make the CHAMP

assay more understandable for readers

o Fig. 2b and 2d: the distribution of the fluorescence signal (raw data) should be showed for each chase; do all the chase variants provide the same fluorescence peak width during flow cytometry?

- 50 μM of chase compound seems extremely high compared to the 5 μM utilized in CAPA experiments; this difference should be noted and explained (does the chase enter cells more poorly than chloroalkane chases (TMRcl)? Is the az-DBCO reaction slower than the chloroalkane and HaloTag binding?)
- Fig. S6 and S7: these plots appeared to be swapped. This should be corrected to avoid confusion.

Figure 3 and related text:

- Design of a high-throughput pipeline with the CHAMP assay could be very powerful
- Differences in binding of each azide-tagged compound to the DBCO-HaloTag is very important to detect if present
- Fig 3a: Polystyrene should be corrected to polystyrene
- Do the polystyrene beads contain the same concentration of HaloTag per bead as each cell (on average)? If not, it may not be appropriate to compare fluorescence between the two
- Does the binding between beads and azide-tagged compound simply serve as a 100% binding control? Do all compounds not bind the beads the exact same given enough time? If not, then this difference in reactivity with DBCO should be compensated for with the CHAMP assay (an increase in “permeability” could instead be poor binding to DBCO?)
- Why not utilize the digitonin mentioned later in the paper to measure the influence of the plasma membrane on permeability?
- Fig 3b: the compounds should be labelled in some way. I was confused as to what the difference between the compounds would be.
- Fig 3c: what is the x-axis? There is no label.
- Fig 3d: the various compound characteristics are not described in the legend or the text body. what HBA, HBD, or TPSA are is not obvious to the reader.
- The implications of the Fig. 3d correlations are not discussed in any way in the text body.
- Overall, the figure is very unclear in content and meaning.

Figure 4 and related text:

- Investigation of the influence of small structural changes on compound permeability is very interesting
- Fig 4b: tick marks should be shown for the y-axis; also 1.2 is misleading, implying that some compounds accumulate over 1.0 which does not make sense (more compound accumulating in cell than there is HaloTag-DBCO to bind?)
- Fig. 4b-4e: Correlation plots showing the differences in apparent accumulation with changes in structure would be more useful than the bar plots, on which it is very difficult to tell what differences are resulting in changes in accumulation.
- The presence of so many 1.0 apparent accumulation (saturated signal) compounds is very confusing; are these compounds all reaching the cytoplasm of the cells the exact same? Or are there differences in permeability which the CHAMP assay is not detecting because of saturation. The compounds should be used at a lower concentration or with less incubation time to allow detection of differences.
- How long was each compound incubated with the cells at 50 μM ? This should be stated in the figure legend. If compounds are left too long, the HaloTag-DBCO could be degraded over time.

Figure 5 and related text:

- Dead cells should be accounted for with this experiment; 24 hours with a peptide at 50 μM could easily lead to toxicity and misleading measurements.
- Fig 5c: why is the R5az showing an apparent accumulation of 0.6 when it showed a maximum of 0.3 or so in 5a?
- Why use the polyarginine peptides at all when you have sCy5az, since sCy5az is known to enter cells poorly on its own?
- Fig 5d: is the R9az toxic to cells? Could it be that the R9az is making cells more unhealthy and thus endocytose less peptide? With 24 hours of a D-amino acid peptide at 50 μM (cells cannot degrade the D-amino acids as easily), this could be the case.
- Fig 5g: would 36GFP ever show any signal with no az tag? It cannot ever show any apparent accumulation by CHAMP if it does not have the az tag correct? It only makes sense to compare the amount of GFP fluorescence in the cells, not the CHAMP assay if the GFP does not have az.

Figures 6-7:

- Investigation of the influence of peptide cyclization on cell permeability is a great demonstration of the CHAMP assay
- As with Figure 4, why not include correlation plots (R-squared values or a similar measurement) to show how these peptide modifications impact cell permeability? Current structure is very difficult to read (reader must look at bar plot, then compounds, then back and forth to understand the meaning of the bar plot).

Reviewer #2

(Remarks to the Author)

The extent and rate at which molecules accumulate inside the cytosol is a critical parameter in drug development. However, obtaining this information is analytically challenging. In the context of peptides and proteins fluorescent tags have been widely employed for this purpose, however, due to their size, fluorophores may greatly affect the molecular behavior of the molecule of interest, in particular for smaller molecules.

The system is based on HeLa cells stably expressing the HaloTag protein. These cells are then incubated with a heterobifunctional DBCO-cl linker which penetrates the plasma membrane and then labels the Halo-tag inside the cell. Then, cells are incubated with the molecules of interest, carrying an azide tag (pulse). Following this step, cells are chased with an TMRcl fluorescent probe which binds to the unreacted DBCO sites. Thus, the higher the cellular fluorescence after the chase, the less the molecules of interest had penetrated the cells. To correct for differences in azide reactivity, molecules of interest are also reacted with DBCO-functionalized beads for which reaction is not impeded by the plasma membrane.

The final measure of permeability is the “apparent accumulation”. The approach is then applied to a library of small molecules, to cell penetrating peptides, other peptide scaffolds and supercharged GFP.

The concept is interesting, and there definitely is a need for analytical methods to determine membrane permeation. However, in my view, more effort should have been invested into a more careful validation of the approach rather than a broad application that leaves many open questions, also due to poorly chosen experimental parameters.

In my view, the manuscript may be suitable for publication but only after major revisions, requiring several additional experiments and a more thorough comparison against published data.

Figure 2: A set of fluorophores is tested for their azide-mediated intracellular accumulation, with TMRaz showing the highest signal and least background in cells not expressing the HaloTag. Concentrations of dyes were high (50 microM). I have difficulties to reconcile these findings with my own experience. The results imply that the dyes nearly instantaneously distribute freely across the plasma membrane in spite of their size, hydrophobicity and charge. In fact, Figure S6 shows that already after 5 min, TMRaz has fully saturated the binding sites. Not without reason, fluorescein-diacetate is used to load cells with the dye which, upon hydrolysis is captured inside the cells – a result that is in stark contrast to the findings presented here. The Cy-dyes are sulfonated, and thus also poorly membrane permeable. TMR accumulates in mitochondria, thus I have problems to explain the low background for wt HeLa cells.

Experiments need to be carried out in which the dyes are incubated with the cells in the presence of propidium iodide to establish that the plasma membrane stays in fact intact. Also a formazan (MTT) assay should be conducted to demonstrate that the (mitochondrial) viability of cells is maintained.

In addition, it should be established to which degree the HaloTag binding sites are saturated, using Western-Blot for the HaloTag. If the DBCO by itself does not induce a large enough shift in apparent MW, then cells could be incubated in one of the peptides used in one of the later experiments.

This experiment is also crucial in light of the data shown in Figure S4, namely that also at the highest concentration of DBCO tested, labeling with TMRcl still clearly stays above background.

Figure 3: There is no information to be gained from this experiment – apart from a demonstration that the assay could be conducted in a high-throughput format. However, the whole story could gain much more from a thorough cross validation against a rather small set of compounds for which information on membrane permeability is available. Ideally, for the major part of compounds, a correlation was found. I thus propose to conduct such a set of experiments.

Figure 4: All compounds were tested at a concentration of 50 microM. Information on incubation time is missing in the legend. I have the same concern on relevance as for the assay as a whole and strongly advise to also include lower concentrations.

Figure 5: The experiments with R9/r9az also raise doubts about the validity of results. At concentrations > 20 microM both CPPs start showing toxicity, in particular over a time frame of 24 h. In addition, at 50 microM, both peptides show rapid cytosolic entry and strongly accumulate inside the cytosol, already after minutes. I then wonder, how under such extreme conditions, there can be a difference between R9az and r9az at all. Also, I wonder why full accumulation requires 24 h.

The CPPs may present a useful tool to better understand the assay. I propose to incubate cells with Fluo-CPPaz and correlate HaloTag conjugation to cytosolic staining for different incubation times and concentrations. In fact, figure 5b reproduces some key observations on CPPs, namely also the higher cytosolic accumulation at concentrations > 10 microM due to direct membrane permeation.

Figure 5c: Which peptide concentrations were used for which incubation time?

Figure 5g: The results suggest that cytosolic entry of the supercharged GFP is reduced by the plasma membrane by 50 % relative to free binding to beads. Again, this finding contradicts literature. Supercharged proteins were abandoned as a strategy for protein delivery due to poor effectiveness. Also, it has been shown that only a fraction of cellular protein reaches the cytosol, the major part being captured in endosomes. I recommend a critical referencing and comparison to 10.1038/s41598-017-13469-y which has provided a thorough quantitative assessment of cytosolic delivery.

Minor points:

In all legends, concentrations and incubation times should be specified as these are main experimental parameters.

Why is there so much more staining in figure 2f in comparison to figure 1s? Also, I assume that the Red channel in 2f should be labeled TMRaz. Authors should critically re-examine the first part of the manuscript. In my idea, there are some misuses of az versus cl.

Figure S3: The concentration range is not well suited to establish the concentration dependence of the reaction as already at the lowest tested concentration there is 70 % of the maximal signal.

Figure 5b – correct: cells were not pulsed with 50 microM peptide but with peptide at the different concentrations.

Figures S5 and S6 are swapped.

Last sentence of section on antibiotics testing: correct

Cytoplasm and cytosol seem to be used synonymously, when they are not.

Legend to figure 1: correct last two sentences

Reviewer #3

(Remarks to the Author)

In this study, authors developed an assay to measure the cellular membrane permeability of small molecules, peptides, and proteins. Previously, the Kodadek and Kritzer groups reported assays for assessing peptide permeability. A major drawback of their methodologies was the large size of the tags required for permeability measurements. In contrast, the present study employs an azide group as a minimal tag for assessing membrane permeability.

However, a very similar assay, termed NanoClick, was previously reported (ACS Chem. Biol. 2021, 16, 293-309). Therefore, the assay described in this manuscript lacks novelty. As such, the manuscript in its current form is not suitable for publication. The authors may consider rewriting the manuscript with the primary focus on the permeability data obtained and discussing the structure-permeability relationships based on these results.

Below are some minor comments regarding the results and discussion:

1. In the section "Screening of Azide-Modified Small Molecule Compounds", the authors state: "A similar trend was observed: amidated 2p accumulated more efficiently than 2n, although both compounds showed reduced accumulation relative to scaffold 1, likely due to increased molecular size (Fig. 4b)." However, according to Fig. 4b, the observed trend appears to be the opposite. Either the description or the figure labeling may be incorrect and should be corrected.
2. In the same section, the authors state: "Among the antibiotics with similar reactivity, our results revealed that other than three antibiotics – purAZ1, zolAZ3, and sulAZ1 – showed significantly higher fluorescence signals, indicating more cytosolic accumulation (Fig. 4g)." In this assay, higher fluorescence signals should indicate lower cytosolic accumulation. Therefore, the statement may need to be revised to: "lower fluorescence signals".
3. Regarding the evaluation of peptides, there are concerns that peptide degradation may be affecting the results. In all experiments involving peptides, the authors should confirm that the peptides remain stable during the permeability assays.
4. The structures of Nalk series are missing from Figure 7 and Fig. S12. These structures should be included in the respective figures.

Version 1:

Reviewer comments:

Reviewer #1

(Remarks to the Author)

The authors have adequately addressed my concerns in their revision.

Reviewer #2

(Remarks to the Author)

The authors took a serious effort to answer the reviewers' comments and the clarity of the manuscript has improved considerably. I also appreciate the authors' willingness to pick up on synthesizing the fluorescently labelled peptide (Figure S17) If I understand correctly, the left panel shows fluorescence from the fluorescein moiety of the peptide whereas the right panel shows TMR fluorescence as a result of the CHAMPS assay. As expected both are largely inversely correlated. As I would expect, TMR fluorescence decreases with peptide concentration, even though it is not clear why it slightly increases with peptide incubation time. For clarity, I would label the Y axes with MFI fluorescein (for A) and MFI TMR (for B). It would also help, if the authors explicitly mentioned where there are deviations from expectations, as for the 50 microM peptide were for peptide fluorescence, there is a much larger dependence on incubation time than for TMR fluorescence.

I agree with reviewer 3 that the explanation of the accumulation of antibiotics is not clear and in this case the reply of the authors did not help to clarify. The authors need to look into this point again and rephrase.

The adjustments proposed above are minor. The answers to all comments provided by the authors are of high quality, I therefore support publication of this manuscript after minor final adjustments.

Reviewer #3

(Remarks to the Author)

The authors have addressed all of my concerns raised in the initial review. However, there are still some shortcomings in the revised points, as outlined below:

1. In response to one of my comments, the authors cited and introduced the NanoClick assay in the introduction. However,

the discussion regarding the advantages and disadvantages of NanoClick assay relative to the CHAMP assay, which was mentioned in the point-by-point response document, has not been incorporated into the revised manuscript. It would be desirable to include such a comparative discussion between NanoClick and CHAMP to help readers better understand the differences between these assays.

2. In response to another comment, the authors stated in the point-by-point response document that they added the following discussion: "We note that, as with other cellular accumulation methods (including LC–MS/MS, NanoClick, and CAPA), molecules of interest may undergo degradation, resulting in an apparent total signal that differs from that of the intact parent compound. This limitation is inherent to these assay formats. Users should therefore consider assessing whether, and to what extent, degradation occurs in their system, as this may influence the interpretation of the biological readout." However, it appears that this discussion section is not included in the revised manuscript. Please review the revised manuscript and include the discussion.

Dear Dr., Huijuan Guo

We thank you for your time and effort on this submission and similarly of all the reviewers. Please see below our point-by-point responses in blue.

Responses:

Reviewer #1

Comments:

This is a paper that represents a potentially valuable improvement of the CAPA assay to measure cell penetration. Briefly, this assay involves an azide label on cell penetrating agents, as opposed to a chloro tag. The azide is smaller than the chlorotag. Conceptually, this likely means that the tag interferes less with the cell penetrating agents. This is likely for small penetrating agents, less likely for larger molecules. To increase potential impact, the authors should certainly test the effect of Azide vs Chlorotag on the cell penetration of a model compounds. For large molecules like proteins, an azide could be useful because it can be integrated with unnatural amino acid incorporation. One of the downside of the approach is the need for cell pretreatment with a intermediate step that prime the Halotag reporter. The effect of this treatment on cell penetration is lacking in the study. This is critical as the claims on cell penetration depends on this step not modifying membranes at all.

Part of the paper is to establish the technology; part is to learn about cell penetration. The paper generally lacks technical details that would convincingly demonstrate that cell penetration does not result from artifacts (e.g. cell death, effect of DBCOcl treatment - see below).

Figure 1 and related text

The CHAMP assay is an excellent idea to minimize the size of the chloroalkane on probes with CAPA experiments; an azide instead of a chloroalkane definitely appears to be less problematic for the compound being studied.

There is no example of the raw data generated by the CHAMP assay in this figure, which can make it confusing to understand what is being measured. It would be helpful to show an example of the changing fluorescence of the azide chase from flow cytometry or SDS-PAGE with low or high permeability compounds. Also, how quantitative measurements of cell penetration are derived from flow cytometry should be presented, along with associated statistical analysis.

- We thank the reviewer for this comment. We have modified Figure 1 to better align with the suggestion. Because SDS-PAGE is not used in our analyses, we instead focused on presenting representative raw flow-cytometry data for a set of molecules.

Is it possible that DBCOcl alters the permeability of cells in the CHAMP assay at all? In the CAPA experiment, the chlorotag chase is done only after incubation with the compound being tested. However, the CHAMP assay requires DBCOcl treatment prior to compound treatment. The potential influence of this pre-treatment should be tested. Different experiments use different wait times between DBCOcl and penetration assays. Hence, the potential effects of DPCOcl should be tested for all these conditions.

- We thank the reviewer for this insightful comment. We agree that different experiments employ different conditions, and therefore the potential influence of DBCOcl on cell permeability was carefully considered. As shown in Figures S5 and S7, concentration- and time-dependent optimization of DBCOcl treatment was performed to identify conditions that minimize any unintended membrane effects. Furthermore, the possible impact of DBCOcl on cellular permeability was directly evaluated using a Propidium Iodide (PI) uptake assay, a standard method for assessing membrane integrity as shown in Supplementary Figure S8.

The process of synthesis of compounds or peptides with an azide tag (include) is not discussed at all; this detail is crucial for the CHAMP assay. Are azides easy or limited with attachment to compounds? If this process is easy, this should be explained as a strength of the assay. If it is a limitation, that should be explained.

- We thank the reviewer for pointing this out. Most azides are known to be chemically stable, and a commercial library of 380 well-defined azide-containing small molecules was procured from Enamine (catalog no. AZD-380-X-100). The azido-peptides synthesized in our laboratory were purified by HPLC and characterized by mass spectrometry. Detailed synthetic procedures for these azido-peptides have now been included in Supplementary Information.

- S17 (Methods):

- o What buffer are the DBCOCl, azide-tagged compounds, and TMRaz in during incubation with cells? Is PBS being used? Or perhaps L-15 buffer? This should be noted.

- We thank the reviewer for this comment. Stock solutions of DBCOCl, azide-tagged compounds, and TMRaz were prepared in DMSO (50 μ M) and subsequently diluted in cell culture medium to achieve the desired working concentrations. This information has now been included in the Supplementary Information (page S28, Methods section).

Are dead cells removed from analysis at all? Could it be that dead cells lead to a misleading increase in permeability?

- We thank the reviewer for this comment. Yes, non-adherent dead cells were removed by washing the wells three times prior to fixation. Only the live, adherent cells were fixed at the end of the assay and subsequently analyzed by flow cytometry. Therefore, the likelihood of dead cells contributing to any misleading assessment of permeability is minimal.

Figure 2 and related text:

- Testing the different compounds for the az chase is an essential step for the CHAMP assay
- Background binding of the az chase to cells (non-HaloTag cells) appears very low
- We thank reviewer for this comment. Indeed, we found that several dyes exhibited low background fluorescence, providing a robust signal-to-noise window for measuring apparent accumulation.
- Could this CHAMP approach be validated with a regular CAPA experiment. This could be one way to show that the CHAMP assay is consistent with previous strategies and that the DBCOCl treatment is not altering the results
- We thank reviewer for this comment. In fact, we did perform a traditional CAPA assay with DBCOCl compound (found in Figure S10). There was a concentration dependent change in cellular fluorescence with increasing levels of DBCOCl.
- More raw output data from the chase should be shown (either from flow cytometry or SDS-PAGE) to make the CHAMP assay more understandable for readers.
- We thank reviewer for this comment. The raw output data for TMRaz chase is shown in supplementary information under Miscellaneous data: Fig 1, and Fig 2.

- o Fig. 2b and 2d: the distribution of the fluorescence signal should be showed for each chase; do all the chase variants provide the same fluorescence peak width during flow cytometry?

- We thank reviewer for this comment. The distribution of fluorescence signal is shown for each chase variant and placed on the supplementary information under miscellaneous data Fig.3.
- 50 μM of chase compound seems extremely high compared to the 5 μM utilized in CAPA experiments; this difference should be noted and explained (does the chase enter cells more poorly than chloroalkane chases (TMRcl)? Is the az-DBCO reaction slower than the chloroalkane and HaloTag binding?)
- We thank the reviewer for this comment. We have not directly compared 5 μM and 50 μM concentrations of TMRcl. However, as shown in Figure S9, when using the most effective chase molecule from our azide panel, a concentration of 50 μM was sufficient to fully occupy the DBCOcl landmark within the cell. Based on these observations, 50 μM was selected as the optimal concentration for the CHAMP assay.
- Fig. S6 and S7: these plots appeared to be swapped. This should be corrected to avoid confusion.
- We thank reviewer for this comment. The plots are corrected with appropriate figure captions.

Figure 3 and related text:

- Design of a high-throughput pipeline with the CHAMP assay could be very powerful
- Differences in binding of each azide-tagged compound to the DBCO-HaloTag is very important to detect if present.
- We appreciate that there may be differences similar to those observed with other covalent or self-labeling strategies. For example, in the CAPA assay, it is possible that chloroalkane-tagged molecules differ in their binding to HaloTag. Despite these potential challenges, CAPA is now widely used both academically and industrially because it provides subcellular localization and resolution of molecular arrival in a way that had not been previously achievable, making it a critical tool for drug discovery.
- Fig 3a: Polysterene should be corrected to polystyrene
- We thank reviewer for this comment. Polysterene has been corrected to polystyrene.
- Do the polystyrene beads contain the same concentration of HaloTag per bead as each cell (on average)? If not, it may not be appropriate to compare fluorescence between the two.
- We thank the reviewer for this comment. The beads are functionalized with DBCO but lack a HaloTag label; therefore, a direct comparison of fluorophore labeling between the beads and cells is not possible. Instead, we compared the reaction kinetics between DBCO and the azide-containing molecules, as this interaction is independent of HaloTag in both systems. For this reason, the beads were used to evaluate how differences in DBCO–azide reactivity influence the observed assay readout.
- Does the binding between beads and azide-tagged compound simply serve as a 100% binding control? Do all compounds not bind the beads the exact same given enough time? If not, then this difference in reactivity with DBCO should be compensated for with the CHAMP assay (an increase in “permeability” could instead be poor binding to DBCO?)

- We thank the reviewer for this comment. The reactivity of DBCO with all azides remains consistent on the beads across both concentration- and time-dependent conditions. Therefore, any differences in the cellular assay cannot be attributed to variations in DBCO–azide reactivity. This allows the CHAMP assay to specifically measure the permeability of compounds across the cell membrane, with the reactivity factor effectively controlled for.
- Why not utilize the digitonin mentioned later in the paper to measure the influence of the plasma membrane on permeability?
- We thank the reviewer for this comment. The effect of digitonin has already been described in the manuscript and is presented in Supplementary Information Figure S15.
- Fig 3b: the compounds should be labelled in some way. I was confused as to what the difference between the compounds would be.
- We thank reviewer for this comment. The figure has been labelled to show differences between the compounds.
- Fig 3c: what is the x-axis? There is no label.
- We thank reviewer for this comment. The X-axis has been labelled.
- Fig 3d: the various compound characteristics are not described in the legend or the text body. What HBA, HBD, or TPSA are is not obvious to the reader.
- We thank reviewer for this comment. The abbreviations of HBA, HBD, or TPSA have been included in the main text.
- The implications of the Fig. 3d correlations are not discussed in any way in the text body.
- Overall, the figure is very unclear in content and meaning.
- We thank reviewer for this comment. The goal of this portion of the study was simply to demonstrate the power in analyzing accumulation at this scale. Your point in terms of content/meaning was addressed by adding the following text to the section “For any given data point, the plot shows a specific molecule. When the whole-cell fold change is lower than the corresponding bead-based fold change, it suggests that the occupancy of the molecule on DBCO landmarks is impeded by the membrane of the cell. This comparison allows us to isolate and assess the impact of the membrane on each compound and its intrinsic reactivity with DBCO.”

Figure 4 and related text:

- Investigation of the influence of small structural changes on compound permeability is very interesting.
- Fig 4b: tick marks should be shown for the y-axis; also 1.2 is misleading, implying that some compounds accumulate over 1.0 which does not make sense (more compound accumulating in cell than there is HaloTag-DBCO to bind?).
- We thank reviewer for this comment. The more detailed tick marks are added for the figures 4b, 4d, and 4e.

- Fig. 4b-4e: Correlation plots showing the differences in apparent accumulation with changes in structure would be more useful than the bar plots, on which it is very difficult to tell what differences are resulting in changes in accumulation.

- We thank reviewer for this comment. We do not believe there is sufficient data volume to support correlational analyses such as principal component analysis (PCA), if that is what the reviewer intended. Given the limited dataset and the heterogeneity of the modifications, a correlation analysis would not reliably capture relationships between these structural changes and their impact on accumulation.

- The presence of so many 1.0 apparent accumulation (saturated signal) compounds is very confusing; are these compounds all reaching the cytoplasm of the cells the exact same? Or are there differences in permeability which the CHAMP assay is not detecting because of saturation. The compounds should be used at a lower concentration or with less incubation time to allow detection of differences.

- We thank the reviewer for this comment. An apparent accumulation value of 1.0 indicates that those compounds are effectively reaching the cytosol. The reviewer is correct that, similar to CAPA, distinguishing differences within a series of compounds may require refined conditions. In this case, reducing the concentration or shortening the incubation time could provide a clearer window to resolve relative differences. We chose to maintain consistent conditions across all peptidic-small molecule series to facilitate broader comparisons, rather than focusing solely on differences within individual subseries. To address the reviewer's concern, we repeated the experiment for series 1–3 under lower concentration conditions, and the resulting data are presented in Figure S12.

- How long was each compound incubated with the cells at 50 μM ? This should be stated in the figure legend. If compounds are left too long, the HaloTag-DBCO could be degraded over time.

- We thank the reviewer for this comment. The incubation time for each compound at a concentration of 50 μM has been specified in the manuscript. The time-dependent incubation of TMRaz with cells, up to 30 minutes at 50 μM , is shown in Figure S10, which demonstrates that the HaloTag-DBCO remains stable and does not undergo degradation under these conditions.

Figure 5 and related text:

- Dead cells should be accounted for with this experiment; 24 hours with a peptide at 50 μM could easily lead to toxicity and misleading measurements.

- We thank reviewer for this comment. Our new data set of experiments for CPPs uses 25 μM of peptides at 4 hours incubation. Thus, we considered doing cell viability test under this condition for the CPPs as shown in Figure S14. Also, the non-adherent dead cells were removed by washing the wells three times prior to fixation. Only the live, adherent cells were fixed at the end of the assay and subsequently analyzed by flow cytometry.

- Fig 5c: why is the R5az showing an apparent accumulation of 0.6 when it showed a maximum of 0.3 or so in 5a?

- We thank the reviewer for this comment. The data in Figure 5c had been inadvertently misinterpreted. The figure has now been corrected and updated with a new set of data reflecting the accurate analysis.

- Why use the polyarginine peptides at all when you have sCy5az, since sCy5az is known to enter cells poorly on its own?
- We thank the reviewer for this comment. The use of digitonin with sCy5az is also presented in Supplementary Figure S15. In this experiment, R5az was selected as a representative compound from the arginine series, as it exhibits relatively low intrinsic accumulation. This allowed us to assess the extent to which digitonin treatment enhances subsequent accumulation, thereby providing a direct comparison within the arginine-based analogs.
- Fig 5d: is the R9az toxic to cells? Could it be that the r9az is making cells more unhealthy and thus endocytose less peptide? With 24 hours of a D-amino acid peptide at 50 μ M (cells cannot degrade the D-amino acids as easily), this could be the case.
- We thank the reviewer for this comment. A 24-hour incubation of cell-penetrating peptides (CPPs) can indeed be toxic to cells. In our updated experiments, we used 25 μ M concentration of CPPs with a 4-hour incubation period. We performed a cell viability assay under these conditions, and the results are shown in Supplementary Figure S18.
- Fig 5g: would 36GFP ever show any signal with no az tag? It cannot ever show any apparent accumulation by CHAMP if it does not have the az tag correct? It only makes sense to compare the amount of GFP fluorescence in the cells, not the CHAMP assay if the GFP does not have az.
- We thank reviewer for this comment. The reviewer is correct in describing that for the supercharged proteins, it should be possible that the fluorescence of the GFP itself can be monitored (similar to the direct coumarin-based read out) and it should correct with the CHAMP results. To this end, we performed the assay again and monitored the GFP-based fluorescence signal (not CHAMP). This data is now included in Fig. S20. Satisfyingly, we found that the direct fluorescence signals from GFP correlated with the CHAMP based results in which the

Figures 6-7:

- Investigation of the influence of peptide cyclization on cell permeability is a great demonstration of the CHAMP assay
- We thank the reviewer for this comment.
- As with Figure 4, why not include correlation plots (R-squared values or a similar measurement) to show how these peptide modifications impact cell permeability? Current structure is very difficult to read (reader must look at bar plot, then compounds, then back and forth to understand the meaning of the bar plot).
- We thank the reviewer for this helpful suggestion. In this case, the peptide modifications are structurally diverse rather than following an incremental pattern. Therefore, there is a lack of a meaningful continuous variable against which to generate a conventional correlation plot. A correlation analysis would therefore not accurately capture the relationship between these structural changes and accumulation.

Reviewer #2

(Remarks to the Author):

The extent and rate at which molecules accumulate inside the cytosol is a critical parameter in drug development. However, obtaining this information is analytically challenging. In the context of peptides and proteins fluorescent tags have been widely employed for this purpose, however, due to their size, fluorophores may greatly affect the molecular behavior of the molecule of interest, in particular for smaller molecules.

The system is based on HeLa cells stably expressing the HaloTag protein. These cells are then incubated with a heterobifunctional DBCO-cl linker which penetrates the plasma membrane and then labels the Halo-tag inside the cell. Then, cells are incubated with the molecules of interest, carrying an azide tag (pulse). Following this step, cells are chased with an TMRcl fluorescent probe which binds to the unreacted DBCO sites. Thus, the higher the cellular fluorescence after the chase, the less the molecules of interest had penetrated the cells. To correct differences in azide reactivity, molecules of interest are also reacted with DBCO-functionalized beads for which reaction is not impeded by the plasma membrane. The final measure of permeability is the “apparent accumulation”. The approach is then applied to a library of small molecules, to cell penetrating peptides, other peptide scaffolds and supercharged GFP.

The concept is interesting, and there definitely is a need for analytical methods to determine membrane permeation. However, in my view, more effort should have been invested into a more careful validation of the approach rather than a broad application that leaves many open questions, also due to poorly chosen experimental parameters.

- We thank reviewer for this comment.

In my view, the manuscript may be suitable for publication but only after major revisions, requiring several additional experiments and a more thorough comparison against published data.

Figure 2: A set of fluorophores is tested for their azide-mediated intracellular accumulation, with TMRaz showing the highest signal and least background in cells not expressing the HaloTag. Concentrations of dyes were high (50 microM). I have difficulties reconciling these findings with my own experience. The results imply that the dyes nearly instantaneously distribute freely across the plasma membrane in spite of their size, hydrophobicity and charge. In fact, Figure S6 shows that already after 5 min, TMRaz has fully saturated the binding sites. Not without reason, fluorescein-diacetate is used to load cells with the dye which, upon hydrolysis is captured inside the cells – a result that is in stark contrast to the findings presented here. The Cy-dyes are sulfonated, and thus also poorly membrane permeable. TMR accumulates in mitochondria, thus I have problems to explain the low background for wt HeLa cells.

- We thank reviewer for this comment. TMRaz does show the background signal in WT HeLa cell, however, due to the high expression of HaloTag protein relatively more amount of TMRaz will be irreversibly accumulated inside the cell.

HaloTag-DBCO that freely diffuses in the cytosol is a molecular sink for the TMRaz. Whereas, TMR is known to localize to the mitochondria, the competing SPAAC reaction with DBCO raises the fluorescence intensity over the mitochondrial association signal for the HaloTag expressing cells. Experiments need to be carried out in which the dyes are incubated with the cells in the presence of propidium iodide to establish that the plasma membrane stays in fact intact. Also a

formazan (MTT) assay should be conducted to demonstrate that the (mitochondrial) viability of cells is maintained.

- We thank the reviewer for this comment. Both TMR and PI dyes emit fluorescence in the same detection channel during flow cytometry and therefore cannot be spectrally separated. To assess cell viability following dye treatment, we performed an MTT assay, the results of which are presented in Supplementary Figure S3.

In addition, it should be established to which degree the HaloTag binding sites are saturated, using Western-Blot for the HaloTag. If the DBCO by itself does not induce a large enough shift in apparent MW, then cells could be incubated in one of the peptides used in one of the later experiments. This experiment is also crucial in light of the data shown in Figure S4, namely that also at the highest concentration of DBCO tested, labeling with TMRcl still clearly stays above background. (cells with TMR and TMRcl to show the background since TMR alone go inside cell and label mito)

- We thank the reviewer for this comment. The saturation of HaloTag binding sites is demonstrated by a gel analysis using one of the most permeable small molecules from our azide library. Corresponding flow cytometry data are also provided to further clarify and support the gel results. This data are presented in Supplementary Figure S11.

Figure 3: There is no information to be gained from this experiment – apart from a demonstration that the assay could be conducted in a high-throughput format. However, the whole story could gain much more from a thorough cross validation against a rather small set of compounds for which information on membrane permeability is available. Ideally, for a major part of compounds, a correlation was found. I thus propose to conduct such a set of experiments.

- We thank the reviewer for this comment. While Figure 3 does not isolate or highlight individual compounds for validation, those data are presented in Figure 4. The purpose of Figure 3 was to demonstrate that our platform is suitable for deployment across large compound libraries, allowing broad patterns of accumulation to be assessed in that context. For the validation experiments, we used small molecules in which specific structural features were systematically varied to generate compounds with expected differences in accumulation. In particular, we evaluated sets of molecules that were neutral, positively charged, or negatively charged. Charge is well established as a major determinant of small-molecule accumulation, and our findings were consistent with those prior observations. We further examined additional structural modifications using peptide-based substrates. However, we acknowledge that larger peptides are inherently more complex and subject to multifactorial parameters that influence overall accumulation profiles.

Figure 4: All compounds were tested at a concentration of 50 microM. Information on incubation time is missing in the legend. I have the same concern on relevance as for the assay as a whole and strongly advise to also include lower concentrations.

- We thank the reviewer for this comment. The information regarding the incubation time has been noted. As suggested, we have now examined the permeability of these peptides at lower concentrations, and the results are presented in Supplementary Figure S12.

Figure 5: The experiments with R9/r9az also raise doubts about the validity of results. At concentrations > 20 microM both CPPs start showing toxicity, in particular over a time frame of 24 h. In addition, at 50 microM, both peptides show rapid cytosolic entry and strongly accumulate inside the cytosol, already after minutes. I then wonder, how under such extreme conditions, there

can be a difference between R9az and r9az at all. Also, I wonder why full accumulation requires 24 h.)

- We thank the reviewer for this comment. To address the concern regarding the use of higher peptide concentrations, we reduced the concentration by half and shortened the incubation time to 4 hours to measure peptide accumulation. The resulting data are presented in Figure 4 of the main text. Additionally, cell viability assays for R9az and r9az were performed under these conditions, and the results are shown in Supplementary Figure S19.

The CPPs may present a useful tool to better understand the assay. I propose to incubate cells with Fluo-CPPaz and correlate HaloTag conjugation to cytosolic staining for different incubation times and concentrations. In fact, figure 5b reproduces some key observations on CPPs, namely also the higher cytosolic accumulation at concentrations > 10 microM due to direct membrane permeation.

- We thank reviewer for this comment. To address the reviewer's concern a new molecule was build that included both the azide tag and a fluorophore for direct analysis. R9LysCoMaz was synthesized and was incubated with cells to correlate HaloTag conjugation to cytosolic staining at two different incubation times and three concentrations. The data is included in Supplementary Figure S17.

Figure 5c: Which peptide concentrations were used for which incubation time?

- We thank the reviewer for this comment. The concentration and incubation time used for the peptide uptake study with digitonin are provided in the figure caption for clarity.

Figure 5g: The results suggest that cytosolic entry of the supercharged GFP is reduced by the plasma membrane by 50 % relative to free binding to beads. Again, this finding contradicts literature. Supercharged proteins were abandoned as a strategy for protein delivery due to poor effectiveness. Also, it has been shown that only a fraction of cellular protein reaches the cytosol, the major part being captured in endosomes. I recommend a critical referencing and comparison to 10.1038/s41598-017-13469-y which has provided a thorough quantitative assessment of cytosolic delivery.

- The bead assay was not performed with the supercharged proteins. As requested, the relevant reference has been added to the main text. In our assay, we do not monitor endosomal localization; it reports solely on cytosolic arrival. Therefore, we have not made any statements or drawn conclusions regarding the fraction of molecules in the endosome relative to the cytosol. The literature indicates that supercharging enhances cytosolic delivery, which is consistent with our observations.

Minor points:

In all legends, concentration and incubation times should be specified as these are main experimental parameters.

- We thank the reviewer for this comment. The concentrations and incubation times have been added to each relevant experimental section for clarity and completeness.

Why is there so much more staining in figure 2f in comparison to figure 1s? Also, I assume that the Red channel in 2f should be labeled TMRaz. Authors should critically re-examine the first part of the manuscript. In my idea, there are some misuses of az versus cl.

- We thank the reviewer for this comment. The increased staining observed in Figure 2f compared to Figure 1s was due to an image processing error. A corrected confocal image for Figure 1s has now been added to the Supplementary Information.

We have gone through the manuscript to remove the confusion between az and cl. Figure S3: The concentration range is not well suited to establish the concentration dependence of the reaction as already at the lowest tested concentration there is 70 % of the maximal signal.

- We thank the reviewer for this comment. A concentration scan of DBCOCl at the lower concentration range has been performed, and the results have been included in the Miscellaneous Data Fig.4

Figure 5b – correct: cells were not pulsed with 50 microM peptide but with peptide at the different concentrations.

- We thank reviewer for this comment. “Cells pulsed with 50 μ M peptide “ is corrected to “ cells were pulsed with peptide at different concentrations”.

Figures S5 and S6 are swapped.

- We thank reviewer for this comment. This has been corrected

Last sentence of section on antibiotics testing: correct

- We thank reviewer for this comment. We have corrected the last sentence.

Cytoplasm and cytosol seem to be used synonymously when they are not.

- We thank reviewer for this comment. Cytoplasm has been replaced with cytosol.

Legend to figure 1: correct last two sentences

- We thank reviewer for this comment. These are now correct.

Reviewer #3

(Remarks to the Author):

In this study, authors developed an assay to measure the cellular membrane permeability of small molecules, peptides, and proteins. Previously, the Kodadek and Kritzer groups reported assays for assessing peptide permeability. A major drawback of their methodologies was the large size of the tags required for permeability measurements. In contrast, the present study employs an azide group as a minimal tag for assessing membrane permeability. However, a very similar assay, termed NanoClick, was previously reported (ACS Chem. Biol. 2021, 16, 293-309). Therefore, the assay described in this manuscript lacks novelty. As such, the manuscript in its current form is not suitable for publication.

The authors may consider rewriting the manuscript with the primary focus on the permeability data obtained and discussing the structure-permeability relationships based on these results.

- We thank reviewer for this comment. The main advantage here is two-fold: first, there is a change from BRET based analysis to fluorescence-based analysis by flow (per cell analysis vs. bulk) and the scale that we performed and benchmarked the assay was several folds higher than the previous NanoClick description. Flow cytometry-based fluorescence assays offer several advantages over plate-reader BRET assays by providing single-cell resolution rather than bulk population averages, allowing you to identify heterogeneous responses and gate on relevant cell subsets. They also offer better control over expression variability, higher signal intensity and dynamic range, and the ability to multiplex with viability dyes or phenotypic markers. This leads to cleaner, more interpretable data and enables linking functional readouts to specific cell populations-capabilities not possible with well-based BRET measurements.

Below are some minor comments regarding the results and discussion:

1. In the section “Screening of Azide-Modified Small Molecule Compounds”, the authors state: “A similar trend was observed: amidated 2p accumulated more efficiently than 2n, although both compounds showed reduced accumulation relative to scaffold 1, likely due to increased molecular size (Fig. 4b).” However, according to Fig. 4b, the observed trend appears to be the opposite. Either the description or the figure labeling may be incorrect and should be corrected.

- We thank reviewer for this comment. The description in the text has been corrected.

2. In the same section, the authors state: “Among the antibiotics with similar reactivity, our results revealed that other than three antibiotics – purAZ1, zolAZ3, and sulAZ1 – showed significantly higher fluorescence signals, indicating more cytosolic accumulation (Fig. 4g).” In this assay, higher fluorescence signals should indicate lower cytosolic accumulation. Therefore, the statement may need to be revised to: “lower fluorescence signals”.

- We thank the reviewer for this comment. The statement “higher fluorescence signal” is correct in the text, as the data presented here are inverse meaning that a higher fluorescence signal corresponds to greater cytosolic accumulation.

3. Regarding the evaluation of peptides, there are concerns that peptide degradation may be affecting the results. In all experiments involving peptides, the authors should confirm that the peptides remain stable during the permeability assays.

- We thank the reviewer for this comment. This consideration is indeed a general feature of all previously described cellular accumulation assays, including LC-MS/MS (where the identities of primary fragments and their respective masses may be unknown), direct dye-conjugation

approaches, and pulse-chase methods such as NanoClick and CAPA. We believe this type of evaluation is necessary for any implementation of the assay. To emphasize this point, we have added a discussion section stating: “We note that, as with other cellular accumulation methods (including LC–MS/MS, NanoClick, and CAPA), molecules of interest may undergo degradation, resulting in an apparent total signal that differs from that of the intact parent compound. This limitation is inherent to these assay formats. Users should therefore consider assessing whether, and to what extent, degradation occurs in their system, as this may influence the interpretation of the biological readout.”

4. The structures of Nalk series are missing from Figure 7 and Fig. S12. These structures should be included in the respective figures.

- We thank the reviewer for catching this. We have revised Figure 7 to include the data for the *N*-methylation library and have added the corresponding structures along with the data for the Nalk series in Figures S21 and S22.

Dear Dr., Huijuan Guo

We thank you for your time and effort on this submission and similarly of all the reviewers again. Please see below our point-by-point response in blue that addresses the remaining concerns of our reviewers.

RESPONSE TO REVIEWERS' COMMENTS:

Reviewer #1:

The authors have adequately addressed my concerns in their revision.

- We thank the reviewer for this comment.

Reviewer #2:

The authors took a serious effort to answer the reviewers' comments and the clarity of the manuscript has improved considerably. I also appreciate the authors' willingness to pick up on synthesizing the fluorescently labelled peptide (Figure S17) If I understand correctly, the left panel shows fluorescence from the fluorescein moiety of the peptide whereas the right panel shows TMR fluorescence as a result of the CHAMPS assay. As expected both are largely inversely correlated. As I would expect, TMR fluorescence decreases with peptide concentration, even though it is not clear why it slightly increases with peptide incubation time. For clarity, I would label the Y axes with MFI fluorescein (for A) and MFI TMR (for B). It would also help, if the authors explicitly mentioned where there are deviations from expectations, as for the 50 microM peptide were for peptide fluorescence, there is a much larger dependence on incubation time than for TMR fluorescence.

- We thank the reviewer for this comment. TMR fluorescence does decrease with the peptide concentration; however, when it comes to increase in the incubation time, a slight increase in signal could be nonspecific peptide binding or the potential that the DBCO landmarks have been exhaustively occupied. This statement has been added to the main text file on the revised manuscript.
- As suggested, for clarity the label on the Y axes has been changed as shown in supplementary figure S17.

I agree with reviewer 3 that the explanation of the accumulation of antibiotics is not clear and in this case the reply of the authors did not help to clarify. The authors need to look into this point again and rephrase.

- We thank the reviewer for this comment. The explanation of antibiotics is rephrased for more clarity in the revised manuscript.

The adjustments proposed above are minor. The answers to all comments provided by the authors are of high quality, I therefore support publication of this manuscript after minor final adjustments.

Reviewer #3:

The authors have addressed all my concerns raised in the initial review. However, there are still some shortcomings in the revised points, as outlined below:

1. In response to one of my comments, the authors cited and introduced the NanoClick assay in the introduction. However, the discussion regarding the advantages and disadvantages of NanoClick assay relative to the CHAMP assay, which was mentioned in the point-by-point response document, has not been incorporated into the revised manuscript. It would be desirable to include such a comparative discussion between NanoClick and CHAMP to help readers better understand the differences between these assays.

- We thank the reviewer for this comment. The discussion regarding the advantages and disadvantages of NanoClick assay relative to the CHAMP assay has been included in the revised manuscript.

2. In response to another comment, the authors stated in the point-by-point response document that they added the following discussion: “We note that, as with other cellular accumulation methods (including LC–MS/MS, NanoClick, and CAPA), molecules of interest may undergo degradation, resulting in an apparent total signal that differs from that of the intact parent compound. This limitation is inherent to these assay formats. Users should therefore consider assessing whether, and to what extent, degradation occurs in their system, as this may influence the interpretation of the biological readout.” However, it appears that this discussion section is not included in the revised manuscript. Please review the revised manuscript and include the discussion.

- We thank the reviewer for this comment. The above-mentioned discussion on the response documents has been added to the revised manuscript.